# Circuit organization of the excitatory sensorimotor loop through hand/forelimb S1 and M1

Naoki Yamawaki[1], Martinna G Raineri Tapies[1], Austin Stults[2], Gregory A Smith[2], Gordon MG Shepherd[1]*

[1]Department of Physiology, Feinberg School of Medicine, Northwestern University, Chicago, United States; [2]Department of Microbiology-Immunology, Feinberg School of Medicine, Northwestern University, Chicago, United States

**Abstract** Sensory-guided limb control relies on communication across sensorimotor loops. For active touch with the hand, the longest loop is the transcortical continuation of ascending pathways, particularly the lemnisco-cortical and corticocortical pathways carrying tactile signals via the cuneate nucleus, ventral posterior lateral (VPL) thalamus, and primary somatosensory (S1) and motor (M1) cortices to reach corticospinal neurons and influence descending activity. We characterized excitatory connectivity along this pathway in the mouse. In the lemnisco-cortical leg, disynaptic cuneate→VPL→S1 connections excited mainly layer (L) 4 neurons. In the corticocortical leg, S1→M1 connections from L2/3 and L5A neurons mainly excited downstream L2/3 neurons, which excite corticospinal neurons. The findings provide a detailed new wiring diagram for the hand/forelimb-related transcortical circuit, delineating a basic but complex set of cell-type-specific feedforward excitatory connections that selectively and extensively engage diverse intratelencephalic projection neurons, thereby polysynaptically linking subcortical somatosensory input to cortical motor output to spinal cord.

*For correspondence:
g-shepherd@northwestern.edu

## Introduction

Functions of the hand and forelimb depend on sensorimotor circuits spanning multiple levels of the central nervous system (*Kleinfeld et al., 2006*; *Arber and Costa, 2018*). At the earliest, most reflexive stage, somatosensory afferents are tightly coupled to motor neurons in the spinal cord. Through a longer loop, somatosensory pathways ascending via brainstem and thalamus reach corticospinal neurons in cortex. The major nodes sequentially traversed in this transcortical pathway include the cuneate nucleus, ventral posterior lateral (VPL) nucleus of thalamus, the hand/forelimb-related primary somatosensory (S1) and motor (M1) cortices. The macroscopic structure of these lemnisco-cortical and corticocortical pathways is well-known from classical anatomy (*Brodal, 1981*) and supported by in vivo electrophysiology (*Andersson, 1995*). However, the cellular-level synaptic connectivity linking the major nodes, whereby peripheral inputs are ultimately conveyed to corticospinal neurons in S1 and/or M1, remains largely uncharacterized for these hand-related circuits. Elucidation of this circuit organization will be an important step toward characterizing basic mechanisms underlying somatosensory-guided control of the hand and forelimb and related aspects of sensorimotor integration in motor cortex (*Hatsopoulos and Suminski, 2011*), and can potentially inform translational approaches to restore hand function in neurological conditions (*Edwards et al., 2019*).

In contrast, much is known about the circuit connections and structure-function relationships in corresponding transcortical pathways in the whisker-barrel system of rats and mice (*Feldmeyer, 2012*; *Feldmeyer et al., 2013*; *Petersen, 2019*; *Staiger and Petersen, 2021*). Similar to the hand-related pathways, the ascending somatosensory pathways in this system include lemniscal and

corticocortical pathways traversing the ventral posterior medial (VPM) nucleus, whisker S1, and whisker M1; additionally, however, a paralemniscal pathway conveys whisking-related signals via the posterior (PO) nucleus. While both systems are used for active sensing, they differ in fundamental ways, ranging from the structure and function of the sensors (actively whisked vibrissal hairs versus glabrous pads and hairy skin) and proprioceptive systems (muscle spindles present in forelimb but largely absent in vibrissal musculature) (*Moore et al., 2015*; *Severson et al., 2019*) to the modes of operation (bilaterally coupled oscillatory whisking versus diverse forelimb movements for manipulation and locomotion). Differences in pathway anatomy may reflect these behavioral specializations; the S1 and M1 areas for the whiskers are widely separated, whereas those for the hand/forelimb are side-by-side, and the primary source of corticocortical input to whisker M1 is the contralateral whisker M1, whereas that for forelimb M1 is the adjacent ipsilateral forelimb S1, suggesting a more prominent role of somatosensory feedback (*Colechio and Alloway, 2009*). With this mix of similarities and differences, the extent to which the organizational features of the whisker-related transcortical circuits pertain to the hand-related circuits is unclear.

Mice offer a favorable model for investigating these hand-related transcortical circuits, as they display a variety of hand and forelimb movements including highly dexterous manipulation behaviors, directional reaching, and more (e.g. *Whishaw et al., 1998*; *Guo et al., 2015*; *Galiñanes et al., 2018*; *Barrett et al., 2020*). Mice have a well-defined hand and forelimb representation in S1, and corticospinal neurons projecting to cells and circuits in the cervical spinal cord feeding into motor neurons innervating forelimb muscles (*Ueno et al., 2018*). Elucidation of hand-related transcortical circuit organization in the mouse could thus provide a valuable comparison both for the rodent whisker-barrel system and the primate hand, and would also facilitate basic research on cortical mechanisms of forelimb functions, for which mice are increasingly used as a model organism.

We used viral labeling, optogenetic photostimulation, whole-cell electrophysiology, and related methods to dissect the cell-type-specific connections in the ascending pathways carrying somatosensory information from the mouse's forelimb, leading to the S1 hand subfield, forelimb M1, and cervically projecting corticospinal neurons. The findings establish a detailed wiring diagram for excitatory somatosensory-to-motor transcortical circuits for the mouse's hand.

## Results

### The S1 hand/forelimb subfield overlaps medially with corticospinal neurons

The overall goal of this study – dissection of the chain of excitatory connections whereby information conveyed by lemnisco-cortical afferents ultimately reaches M1 corticospinal neurons that project back to the cervical spinal circuits controlling the forelimb musculature – entails consideration of the cortical topography involved. The hand-related area of S1 is well-demarcated as a somatotopically organized subfield of the 'barrel map' defined by layer (L) 4 (*Waters et al., 1995*; *Brecht et al., 2004*). However, the cortical distribution of cervically projecting corticospinal neurons, the key cortical components at the downstream end of the transcortical circuit for the hand, is more complex, centering on forelimb M1 (also termed the caudal forelimb area) but also extending into forelimb S1 (*Li and Waters, 1991*; *Young et al., 2012*). Recent results clarify that the corticospinal neurons in forelimb S1 innervate sensory-related neurons in the cervical cord and, unlike those in M1, are not labeled following injections of retrograde transsynaptic viruses in forelimb muscles (*Ueno et al., 2018*). In light of these anatomical complexities, prior to dissecting the transcortical circuit connections we first assessed the topography of the hand subfield of S1 in the mouse, as defined by the presence of L4 barrel-like structures, in relation to the areal distribution of cervically projecting corticospinal neurons. We targeted those projecting to cervical level 6 (C6) in particular (corticospinal[C6-proj] neurons), as C6 is prominently involved in sensorimotor functions of the hand.

Crossing the L4-specific Scnn1a-Cre driver line with the Ai14 Cre-dependent tdTomato reporter line yielded offspring expressing tdTomato in L4 neurons across S1 (*De la Rossa et al., 2013*; *Sigl-Glöckner et al., 2019*). In flattened brain sections (*Figure 1A*), the hand/forelimb S1 subfield contained barrel-like blobs, arrayed in a pattern closely matching that of the rat, where this pattern has been shown to be somatotopically arranged, corresponding to the digits, pads, and wrist, with the D1 and thenar pad representation situated most lateral (adjacent to the lip and mouth area) and the

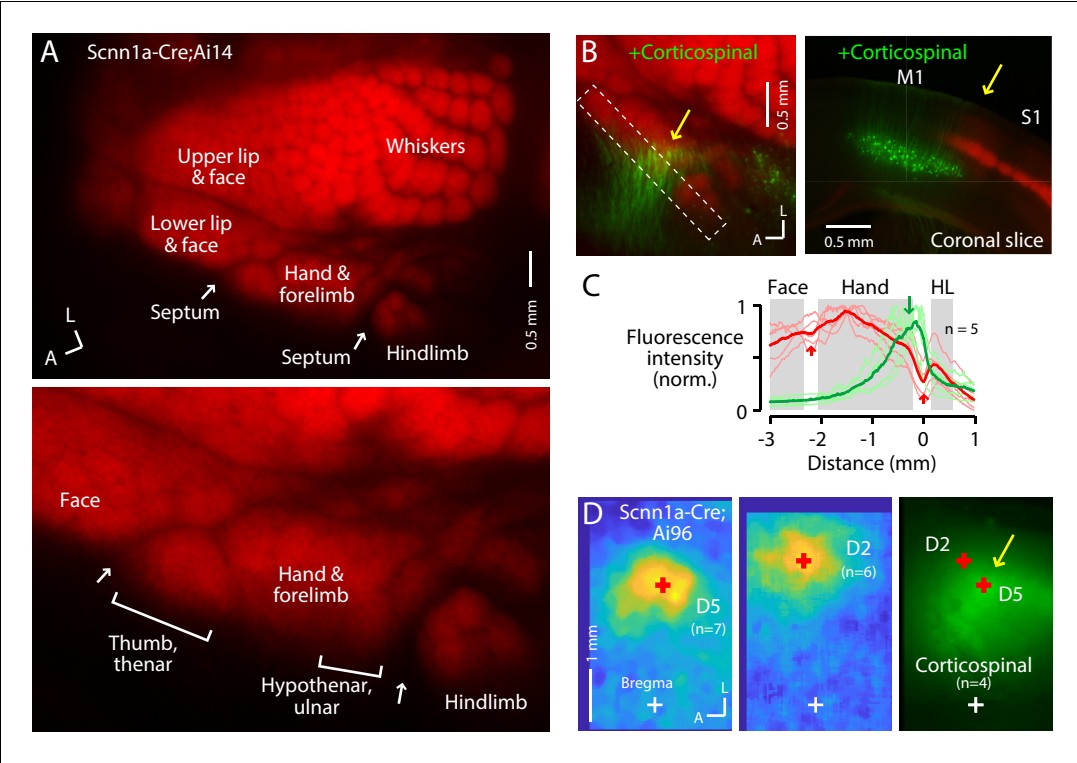

**Figure 1.** The S1 hand/forelimb subfield overlaps medially with corticospinal neurons. (**A**) Top: Flat-mount section through L4 of the cortex of a Scnn1a-Cre;Ai14 mouse, showing the L4 labeling pattern across S1 cortex. This image is of the right side of the brain, as are all dorsal-view images in the figures unless indicated otherwise. A, anterior; L, lateral. Bottom: Enlarged view of the hand region. Septa (arrows) separate the hand region from the neighboring face and hindlimb regions. Labeling of S1 somatotopic subfields is based on prior studies in mice and rats (*Waters et al., 1995*; *Sigl-Glöckner et al., 2019*) and standard atlases (*Dong, 2008*). (**B**) Left: Same, additionally showing corticospinal neurons (green; their dendrites within the section mainly through L4), labeled by cervical injection of AAVretro-GFP. Dashed rectangle: region of interest used to quantify fluorescence profile. Arrow: region of overlap. Right: Coronal section (different animal), showing laminar labeling patterns. This image is of the right side of the brain (midline is to the left), as are all coronal images in the figures unless indicated otherwise. (**C**) Fluorescence intensity profiles across the anteromedial edge of the S1 area (marked by dashed rectangle in image in panel B), for individual animals (lighter traces) and group average (darker, n = 5 animals), showing hand area (gray) bordered by septa (red arrows), with region of corticospinal labeling (green arrow) located medially, in the putative hypothenar/ulnar subregion. Intensity profiles were aligned to the hand-hindlimb septum (x = 0). (**D**) Somatosensory responses mapped by transcranial GCaMP6s imaging in Scnn1a-Cre;Ai96 mice, showing the average responses to stimulation of the fifth (**D5**) and second (**D2**) digits, with the centroids of the responses marked (red '+'), which are also shown superimposed on the average transcranial image of corticospinal labeling, from a subset of the same mice that were injected with AAVretro-GFP in the spinal cord. Maps are aligned to bregma (white '+').

The online version of this article includes the following figure supplement(s) for figure 1:

**Figure supplement 1.** Additional examples and analyses of L4 labeling, corticospinal labeling, and GCaMP imaging.

D5 and hypothenar most medial (adjacent to the hindlimb area) (*Waters et al., 1995*). The mediolateral somatotopic layout of the digits and the cortical magnification of the hand and thumb representations constitute a conserved mammalian pattern found in other rodents such as squirrels (*Sur et al., 1978*) and in monkeys and humans (e.g. *Penfield and Rasmussen, 1950*; *Martuzzi et al., 2014*; *Chand and Jain, 2015*; *Roux et al., 2018*). Septa – linear gaps in the Scnn1a labeling pattern – were found between the hand subfield and neighboring body part representations, and also within the hand subfield, demarcating a lateral region corresponding to the thumb/thenar subregion

(*Waters et al., 1995*); similar septa have been described in monkey S1 as gaps in myelin staining (*Chand and Jain, 2015*).

In the same mice, we retrogradely labeled corticospinal neurons by injecting AAVretro-GFP in the cervical spinal cord at C6. In the cortex, corticospinal[C6-proj] neurons (seen as their proximal apical dendrites in flattened L4 sections) were distributed mostly medial to the hand S1 territory, but with partial overlap at the medial edge of hand S1 (*Figure 1B–D*; *Figure 1—figure supplement 1A*). This region of overlap corresponds to the D5 and hypothenar barrels in the ulnar part of the hand S1 (*Waters et al., 1995*; *Figure 1A*). The corticospinal distribution moreover extended into the relatively large septum between the hand and hindlimb territories of S1, narrowing as it extends posteriorly before merging into a larger cluster of corticospinal neurons situated medial to the posterior medial barrel subfield. Corticospinal labeling was weaker or absent within the hindlimb S1 region itself, and also within the lateral part of the hand subfield corresponding to the D1/thenar subregion. Images of coronal sections gave similar results, confirming that the horizontal distribution of corticospinal neurons, which are located in L5B, extends from M1 into S1, up to ~0.3 mm laterally below the labeled L4 of hand S1 (*Figure 1—figure supplement 1B,C*).

To relate the neuronal labeling patterns to cranial landmarks and stereotaxic coordinates, we imaged the cranium of anesthetized mice to identify the coronal sutures and bregma under brightfield illumination, and transcranially imaged tdTomato fluorescence from L4 neurons and GFP from corticospinal neurons (*Figure 1—figure supplement 1D–G*). Corticospinal labeling was observed in the region commonly identified as forelimb M1, medial to the L4 territory defining S1 (*Ayling et al., 2009*; *Tennant et al., 2011*). However, as noted previously (*Ueno et al., 2018*), the distribution of cervically projecting corticospinal neurons also appeared to extend toward and partially into the medial subregion of hand S1.

To functionally assess if the region of hand S1 overlapping with corticospinal neurons corresponds to the hypothenar/ulnar aspect, we performed somatosensory mapping. First, using CaMKIIa-Cre; GCaMP6s mice to label excitatory cortical neurons, we confirmed the large-scale somatotopic layout of major body part representations in the mouse, with hand S1 situated anterolateral to hindlimb S1, posteromedial to the lower lip and face, and anterior to the vibrissal territory (*Figure 1—figure supplement 1H*), consistent with prior results (*Sigl-Glöckner et al., 2019*; *Guo et al., 2020*). Then, for higher resolution imaging restricted to S1 areas, we used Scnn1a-Cre;GCaMP6s mice to label L4 neurons in S1 areas, which showed that responses to tactile stimulation of the fifth digit (D5) were located in a region corresponding to the posteromedial part of hand S1, in the region of overlap with corticospinal neurons, with the D2 representation located more anterior and lateral (*Figure 1D*, *Figure 1—figure supplement 1I*).

These results, which build on and extend recent characterizations of hand/forelimb-related region of mouse S1 as it relates to the areal distribution of corticospinal[C6-proj] neurons (*Ueno et al., 2018*), demonstrate that the region of partial overlap occurs in a medial part of S1 corresponding to the hypothenar/ulnar subregion of the somatotopic representation of the hand/forelimb area. Subsequently in this study, we generally targeted this subregion of the S1 hand subfield for injections and recordings.

## PRV labeling of the lemnisco-cortical pathway to L4 neurons in S1

As a first step in circuit-tracing, we used pseudorabies viruses (PRV) to anatomically trace the ascending polysynaptic lemnisco-cortical pathway to hand S1. Because L4 neurons are strongly thalamo-recipient in sensory cortex, we targeted them as starter cells for PRV tracing, by injecting the Cre-dependent PRV-Introvert-GFP (*Pomeranz et al., 2017*) into the hand S1 of Scnn1a-Cre mice. After 72 hr (n = 3 mice), Cre-dependent labeling was observed primarily at the injection site in S1, largely restricted to L4 neurons, with additional labeling in a small subregion of the VPL nucleus (*Figure 2A,B*; *Figure 2—figure supplement 1A*). After longer incubation periods (96 hr; n = 3 mice), labeling was stronger at these sites, and also appeared in the cuneate nucleus (*Figure 2C,D*). In these experiments, the precise timing and extent of labeling at different time points may be influenced by multiple factors, such as the time dependence of Cre-mediated gene expression and variable numbers of infected starter neurons in L4 (*Pomeranz et al., 2017*). Furthermore, because the labeling in cortex does not distinguish between first-, second-, and higher order neurons, cortical labeling in L4 presumably represents a mix of these especially at later time points, reflecting the

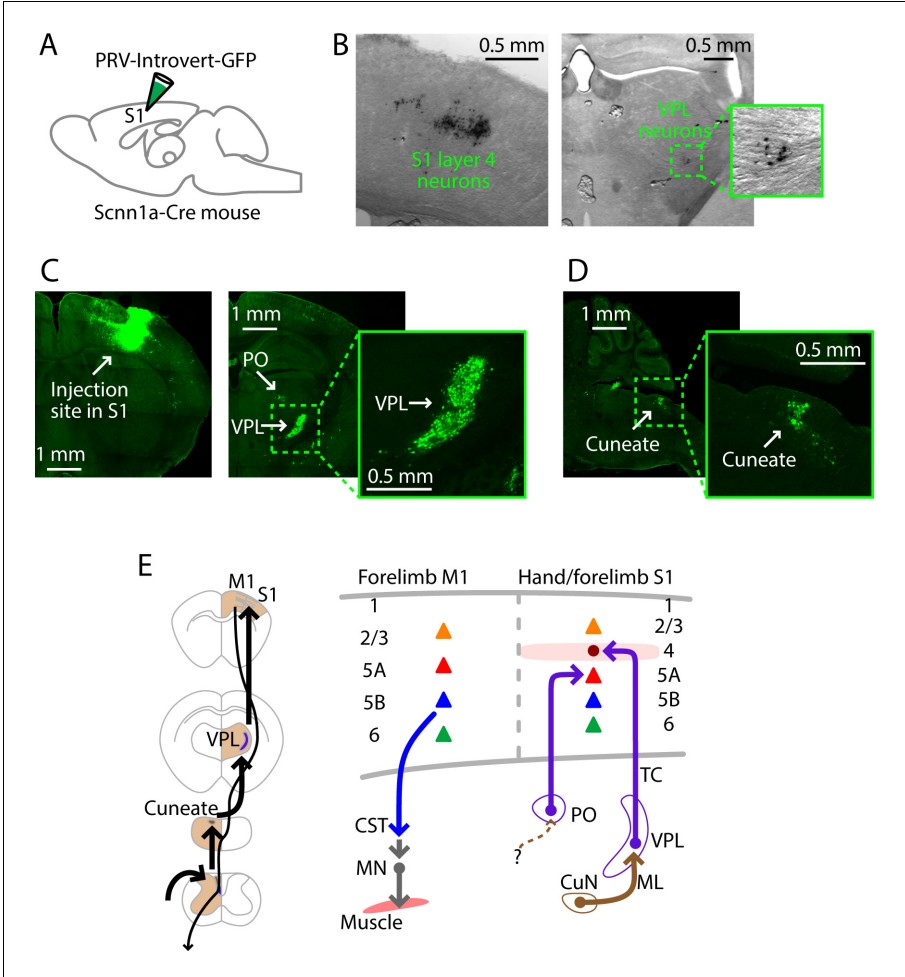

**Figure 2.** PRV labeling of the lemnisco-cortical pathway to L4 neurons in hand S1. (**A**) Schematic depicting the injection strategy. PRV-Introvert-GFP was injected into the S1 hand subfield in Scnn1a-Cre mice, a L4-specific Cre driver line. (**B**) Labeling pattern observed in cortex at the injection site in S1 (left) and in thalamus in VPL (right), after incubation period of 72 hr (coronal slices). PRV labeling was visualized by immunohistochemical amplification of GFP followed by DAB staining. (**C**) Same experiment, but with a longer incubation period of 96 hr. PRV labeling was visualized by immunohistochemical amplification of GFP followed by FITC staining. Left: Coronal slice showing labeling at the injection site in cortex. Right: Coronal slice showing thalamic labeling, at lower (left) and higher (right) magnification. (**D**) Sagittal slice showing cuneate labeling, at lower (left) and higher (right) magnification. (**E**) Schematic summaries depicting the ascending lemnisco-cortical pathway to hand/forelimb S1, via cuneate→VPL→S1-L4 connections, and the descending pathway from forelimb M1 corticospinal neurons. The S1 also receives PO→S1-L5A input.

The online version of this article includes the following figure supplement(s) for figure 2:

**Figure supplement 1.** Additional PRV labeling results.

dense intralaminar interconnectivity in L4 (*Feldmeyer, 2012*). Nevertheless, the timing of the spread to the cuneate suggests a disynaptic lemnisco-cortical circuit (i.e., cuneate→VPL→S1-L4).

In whisker S1, L5A neurons receive paralemniscal inputs from posterior nucleus (PO) neurons, which receive ascending input from a subdivision of the spinal trigeminal nucleus (*Staiger and Petersen, 2021*). We attempted to identify a corresponding cuneo-PO paralemniscal pathway to hand S1 in the mouse by performing the same PRV experiment but with the L5A-specific Tlx3-Cre mouse line. However, 4 days (96 hr; n = 2 mice) after injection of PRV-Introvert-GFP into hand S1, we observed thalamic labeling in PO, but no cuneate labeling (*Figure 2—figure supplement 1B*). As shown previously (*Ueno et al., 2018*), injection of PRV-EGFP into forelimb muscles (biceps)

resulted in labeling (after 72 hr; n = 3 mice) of corticospinal neurons only in forelimb M1, not hand S1 (*Figure 2—figure supplement 1C*).

Collectively, these PRV labeling results provide an anatomical framework of the ascending and descending pathways to guide subsequent electrophysiology-based analysis of the excitatory connections along the transcortical circuits to and through hand-related S1 (*Figure 2E*).

## Cuneate→VPL circuit analysis

Having anatomically traced the cuneate→VPL→S1 pathway by polysynaptic viral labeling, we analyzed each leg of this circuit in more detail, starting with the cuneothalamic pathway. Consistent with the PRV results, injection of retrograde tracer into VPL labeled the cuneate nucleus (n = 3 mice) (*Figure 3A–C*). Injection into PO in the same animals did not label the cuneate but did label the trigeminal nucleus (*Figure 3D*), likely due to spread of tracer into the whisker-related subregion of PO receiving paralemniscal afferents. Similarly, following injection of anterograde tracer into the cuneate and retrograde tracer into S1, in thalamic sections we observed anatomical overlap of cuneate axons and somata of S1-projecting neurons in VPL in a restricted region (n = 6 mice) (*Figure 3E,F*). However, there was often a misalignment in their labeling within VPL, presumably reflecting mismatch in the precise somatotopic representations at the cuneate and S1 injection sites. Cuneate axons were not observed in other thalamic nuclei (e.g. PO, VL), confirming in the mouse that the main ascending cuneothalamic projection is the medial lemniscal pathway to the VPL.

We used optogenetic-electrophysiological methods to characterize excitatory synaptic connectivity in this cuneothalamic circuit. Whole-cell recordings from VPL neurons were made in voltage-clamp mode, with cesium-based intracellular solution containing QX-314 to block sodium channels and action potentials. The ChR2-expressing cuneothalamic axons were photostimulated by brief flashes of light, using wide-field blue LED illumination through a low-power objective. These recordings showed excitatory responses that, although detected in only ~50% of the sampled neurons (n = 11 neurons, out of 23 neurons tested, with response amplitudes more than three times the baseline s.d.), tended to be strong (amplitude −158 ± 64 pA, mean ± s.e.m.) (*Figure 3G,H*). These inputs were blocked by NMDA and AMPA receptor antagonists (1 μM CPP, 10 μM NBQX, n = 3 neurons) and showed short-term depression upon repetitive stimulation (2nd/1st response amplitude: 0.62 ± 0.04; n = 7 neurons, mean ± s.e.m.; p=0.016, sign test) (*Figure 3H,I*). These findings accord with prior results for lemniscal-type inputs to VPM neurons in the whisker-related circuits (*Mo et al., 2017*).

## Cuneate→VPL→S1 circuit analysis

We next sought to characterize the thalamocortical circuits in this pathway, and to do so not just in isolation but as a tandemly connected (i.e., disynaptic) cuneo-thalamo-cortical circuit. We developed a paradigm for this based on AAV-hSyn-Cre for anterograde transneuronal labeling (*Zingg et al., 2017*) to express ChR2 specifically in the cuneo-recipient subset of VPL neurons ([CuN-rec]VPL), together with retrograde tracer injections into either the forelimb M1 or the C6 spinal cord to label projection neurons in S1 (*Figure 4A*). In WT mice, we injected AAV-hSyn-Cre into the cuneate and, to visualize the labeling of cuneate neurons, co-injected AAV-Flex-EGFP, resulting in labeled neurons in the dorsal column nuclei (*Figure 4B*). We additionally injected the VPL, the target of the cuneothalamic projection, with a Cre-dependent AAV-ChR2. This resulted in labeling of [CuN-rec]VPL neurons (*Figure 4C*). In the same slices, we also observed retrogradely labeled VM[M1-proj] neurons as a result of tracer injection into M1. In S1 slices, labeled axons from the [CuN-rec]VPL neurons were seen in L4, along with retrogradely labeled corticocortical L2/3[M1-proj] and L5A[M1-proj] neurons, and corticospinal[C6-proj] neurons in L5B (*Figure 4D*).

This paradigm allowed us, in the same experiment, to concatenate the cuneate→VPL and VPL→S1 stages of the circuit and assess [CuN-rec]VPL input to multiple classes of identified neurons in the cortex. We recorded in S1 slices from L4, corticocortical[M1-proj], and corticospinal[C6-proj] neurons and sampled excitatory currents evoked by photostimulation of the ChR2-expressing VPL axons (*Figure 4E*). Recordings in L4 were targeted to putative excitatory neurons, based on small soma size. We added TTX and 4-AP to the bath solution to isolate monosynaptic responses (*Petreanu et al., 2009*), and again used cesium-based intracellular solution with QX-314 for whole-cell recordings in voltage-clamp mode. We observed a pattern of strongest input to L4 neurons,

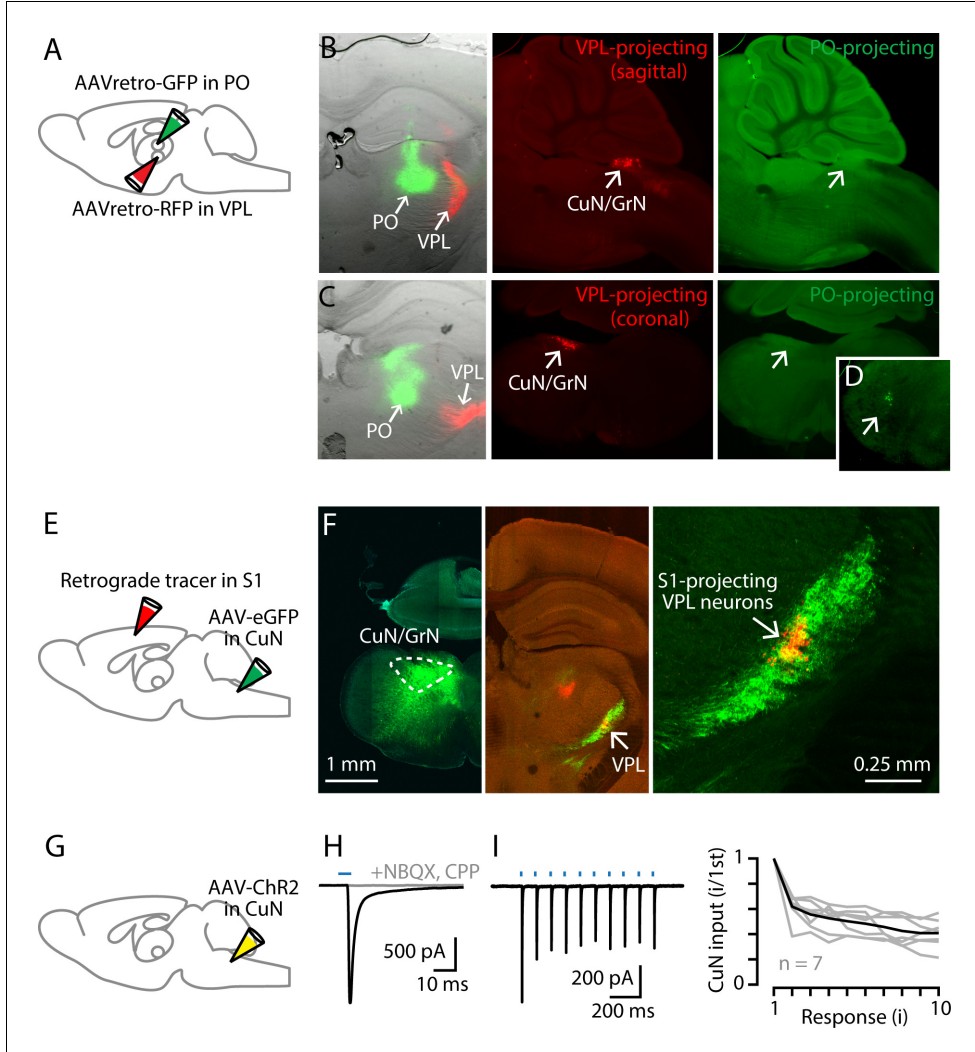

**Figure 3.** Cuneate→VPL circuit analysis. (**A**) Schematic of injection strategy: PO was injected with AAVretro-GFP and VPL was injected with AAVretro-RFP. (**B**) Left: Coronal section showing injection sites in VPL and PO. Middle: Sagittal section showing labeled VPL-projecting neurons in the cuneate nucleus. Right: Same, showing absence of PO-projecting neurons in the same region. (**C**) Same, but with coronal sections, at the level of the right thalamus (left) and left cuneate (middle and right). (**D**) Labeled PO-projecting neurons in the trigeminal nucleus. (**E**) Schematic of injection strategy: forelimb S1 was injected with a retrograde tracer CTB647, and cuneate nucleus was injected with AAV-eGFP. (**F**) Left: Labeling at site of AAV-eGFP injection in the cuneate nucleus (left side of the brainstem). Middle: Labeled cuneothalamic axons in VPL thalamus. Right: VPL$^{S1-proj}$ neurons are situated within the field of labeled cuneothalamic axons. (**G**) Schematic of injection strategy: the cuneate nucleus was injected with AAV-ChR2. (**H**) Example traces showing strong excitatory synaptic responses recorded in a VPL neuron in a thalamic brain slice, evoked by photostimulation of ChR2-expressing cuneothalamic axons. (**I**) Example traces (left) and group data (right) showing strong synaptic depression of responses to trains of photostimuli (amplitude of the $i^{th}$ response divided by that of the first; gray, individual neurons; black, group mean).

moderate-to-low input to L2/3$^{M1-proj}$ and L5A$^{M1-proj}$ neurons, and little or no input to corticospinal$^{C6-proj}$ neurons (n = 9 quadruplets, four mice; p=0.00001, Kruskal-Wallis test) (*Figure 4E*).

We also assessed thalamocortical connectivity by the simpler approach of directly injecting the VPL with AAV-ChR2 (*Figure 4—figure supplement 1A*). Images of the cortical labeling pattern showed, as expected based on the labeling studies described earlier, that the anterogradely labeled VPL axons ramified most densely in L4 of S1, with corticospinal$^{C6-proj}$ neuron distributions found in forelimb M1 with extension into S1 as well, below the barrel-like clusters of VPL axons (*Figure 4— figure supplement 1B*). Electrophysiological recordings in coronal S1 slices showed that responses

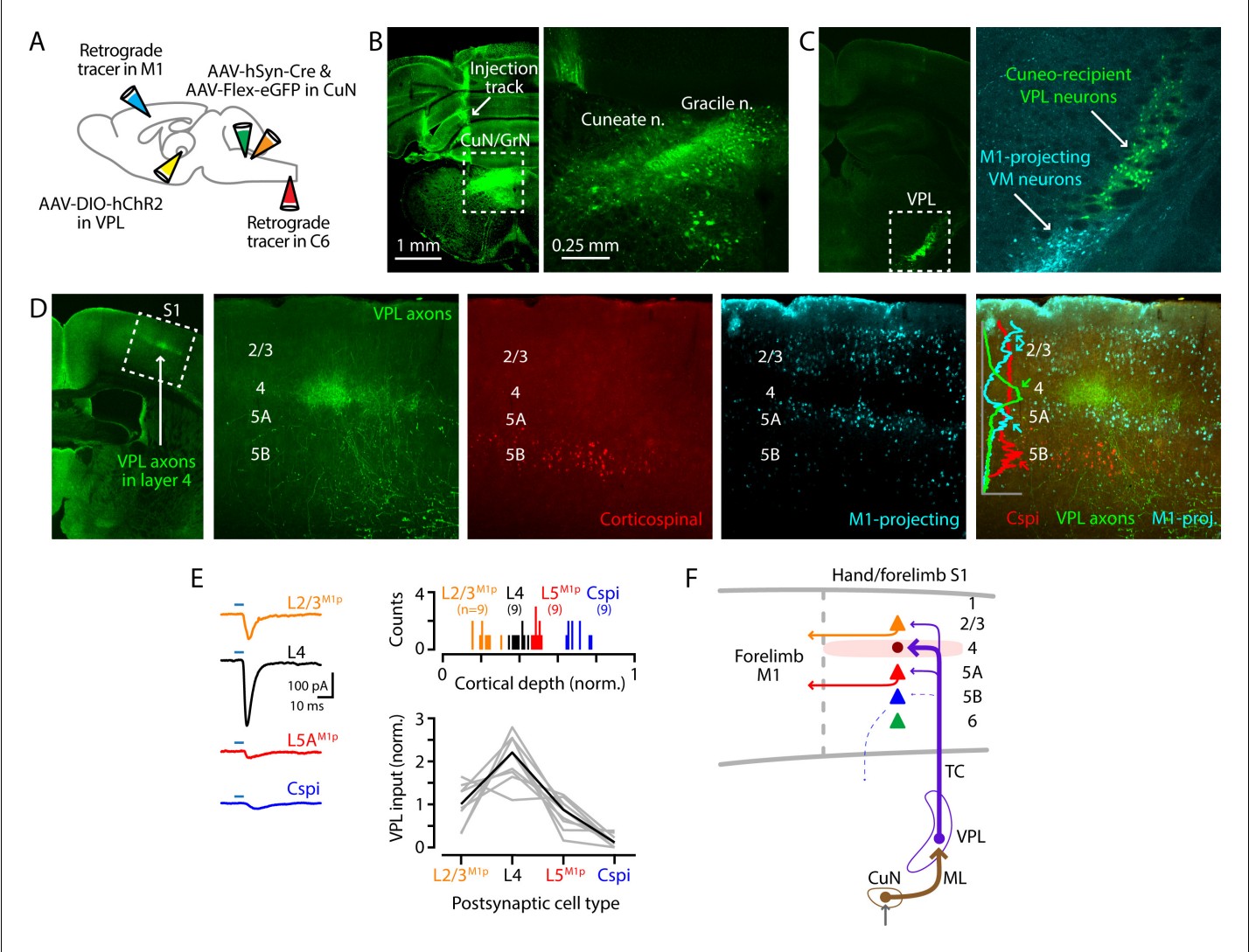

**Figure 4.** Cuneate→VPL→S1 circuit analysis. (**A**) Schematic of injection strategy: forelimb M1 was injected with one retrograde tracer (CTB647) and C6 spinal cord with another (red Retrobeads); the cuneate nucleus was injected with AAV-hSyn-Cre and AAV-Flex-EGFP; and, the VPL was injected with AAV-DIO-hChR2. (**B**) Fluorescence images at low (left) and high (right) power of a coronal section at the level of the dorsal column nuclei, showing labeling in the cuneate and gracile nuclei (left side of the brainstem). Labeling in the latter likely represents spread from the targeted site in the cuneate. (**C**) Coronal section at the level of the VPL nucleus, showing the anterogradely labeled cuneate axons and cuneo-recipient VPL neurons (both in green), along with retrogradely labeled VM[M1-proj] neurons. (**D**) Coronal section at the level of S1, showing the anterogradely labeled VPL axons ramifying in L4 (green; left). The middle three panels show at higher magnification the labeling of VPL axons (green), corticospinal neurons (red, in L5B), and M1-projecting corticocortical neurons (cyan, particularly L2/3 and L5A). These are also shown as a merged image (far right), with a plot of the normalized fluorescence intensity profiles of the different colors. (**E**) Left: example traces of EPSCs evoked by photostimulating ChR2-expressing VPL axons in cortical brain slices, recorded in L2/3[M1-proj], L4, L5A[M1-proj], and corticospinal[C6-proj] neurons in S1. Upper right: Histogram of the normalized cortical depths of each of the S1 cell types sampled. Numbers of cells per group are given in parentheses below the cell type labels. Lower right: Plot of EPSC amplitudes recorded in the four types of postsynaptic S1 neurons. Gray: data from individual sets of four neurons (i.e., sequentially recorded quadruplets). The EPSCs of each quadruplet of recorded neurons were normalized to the quadruplet average. Black: group average, calculated across the set of n = 8 quadruplets. (**F**) Schematic summary of the main findings.

The online version of this article includes the following figure supplement(s) for figure 4:

**Figure supplement 1.** VPL→S1 circuit analysis.

to photostimulation of VPL axons were strongest in L4 neurons, and weaker in corticocortical[M1-proj] (n = 15, 10, and eight for L2/3[M1-proj], L4, and L5A[M1-proj] neurons; eight mice; L2/3[M1-proj] vs L4, p=0.0004; L4 vs L5A[M1-proj], p=0.00005; L2/3[M1-proj] vs L5A[M1-proj], p=0.72; rank-sum test) (***Figure 4—figure supplement 1C–E***). Additional recordings comparing the VPL input to L4 neurons and

corticospinal[C6-proj] neurons showed strong input to the former, and little or no input to the latter (*Figure 4—figure supplement 1F–H*) (n = 8 pairs, two mice; p=0.008, sign-test).

These results thus provide a profile of [CuN-rec]VPL input to hand S1, identifying L4 neurons as the primary targets, with weaker input to corticocortical[M1-proj] neurons in the two adjacent layers and little or no direct excitation of corticospinal[C6-proj] neurons (*Figure 4F*). As previous work has shown strong L4→L2/3 connectivity in local circuits of forelimb S1 of the mouse (*Yamawaki et al., 2014*), the results indicate that those intracortical connections would augment the more direct but lower-amplitude [CuN-rec]VPL→L2/3[M1-proj] connections.

## PO axons mainly excite L5A[M1-proj] neurons in S1

Although the labeling experiments described above did not reveal evidence for a direct afferent pathway from the cuneate to the hand-related subregion of PO (i.e., a counterpart to the whisker-related paralemniscal pathway), the hand subfield of S1 forms cortico-thalamo-cortical circuits with a corresponding subregion of PO through recurrent connections (*Guo et al., 2020*), suggesting that inputs from PO to hand S1 are likely to intersect and interact with lemniscal transcortical circuits, similar to the whisker-barrel system. We therefore dissected PO connectivity to hand S1, by injecting the PO with AAV-ChR2 and the forelimb M1, PO, and/or C6 cervical spinal cord with retrograde tracer(s) (*Figure 5A,B*). The anterogradely labeled PO axons ramified in L1 and L5A (*Figure 5B*). PO inputs were strongest to L5[M1-proj] neurons, weak-to-moderate to L2/3[M1-proj] neurons, and mostly absent to L4 neurons (n = 9, 8, and 9 for L2/3[M1-proj], L4, and L5A[M1-proj] neurons, respectively; 3 mice; L2/3[M1-proj] vs L4, p=0.022; L4 vs L5A[M1-proj], p=0.00004; L2/3[M1-proj] vs L5A[M1-proj], p=0.00004; rank-sum test) (*Figure 5C*). Additional experiments showed stronger inputs to L5A neurons compared to other types of S1 projection neurons, including corticospinal[C6-proj] neurons (n = 9 L5A and 10 corticospinal neurons; 3 mice; p=0.004, sign-test) (*Figure 5D*); L5B[PO-proj] neurons (n = 9 pairs; 4 mice; p=0.004, sign-test) (*Figure 5E*); and, corticothalamic L6[PO-proj] neurons (n = 6 pairs; 2 mice; p=0.031, sign-test) (*Figure 5F*). Thus, collectively these findings (*Figure 5G*) indicate that the main targets of PO projections to hand S1 are L5A neurons, including those forming corticocortical projections to forelimb M1, with additional input to M1-projecting neurons in L2/3 but notably weak or absent input to corticospinal and other major classes of neurons.

## Corticocortical axons from S1 mainly excite L2/3 neurons in M1

To characterize cellular connectivity in the last stage of the circuit leading to M1 and its corticospinal neurons, we used a similar strategy, adapted for cell-type-specific dissection of S1→M1 corticocortical connectivity. Retrograde labeling from M1 demonstrated labeling in S1 mainly of L2/3 and L5A neurons (*Figure 4D*). Focusing on the projection originating from S1 L5A, we used a L5A-specific Cre driver line (Tlx3-Cre) together with stereotaxic injections into S1 of Cre-dependent AAV-ChR2 virus to selectively label the projection from L5A of S1 to M1 (*Figure 6A,B*). Recordings in M1 slices showed that responses to photostimulation of S1 L5A/Tlx3 axons were strongest in L2/3 neurons, and generally either very weak or absent in pyramidal neurons in L5A and L6, and also in corticospinal[C6-proj] neurons (n = 12, 8, 9, and six for L2/3, L5A, corticospinal[C6-proj], and L6 neurons, respectively, recorded as sets of neurons always including L2/3 neurons plus multiple other types; five mice; p=0.00001, Kruskal-Wallis test) (*Figure 6C*). Thus, the L5A-originating component of the S1→M1 corticocortical circuit selectively excites postsynaptic L2/3 neurons (*Figure 6D*).

Similar findings were obtained with shallow injections in S1 that mainly labeled L2/3 neurons (*Aronoff et al., 2010*; *Figure 6E,F*). Again the S1 corticocortical axons primarily excited L2/3 neurons in forelimb M1, with weaker input to L5A neurons and weak or absent input to L5B neurons, including corticospinal neurons (n = 6, 7, 5, 4, and six for L2/3, L5A, unlabeled L5B, corticospinal[C6-proj], and L6 neurons, respectively, recorded as sets of neurons always including L2/3 neurons plus multiple other types; five mice; p=0.0004, Kruskal-Wallis test; corticospinal[C6-proj] is grouped with unlabeled L5B neurons) (*Figure 6G*). Thus, the L2/3-originating component of the S1→M1 corticocortical circuit selectively also excites postsynaptic L2/3 neurons (*Figure 6H*), converging with the L5A-originating component.

These results add key details about the excitatory connectivity in the last stage along the transcortical circuit leading to M1, showing that the main recipients of S1 corticocortical input are L2/3 pyramidal neurons.

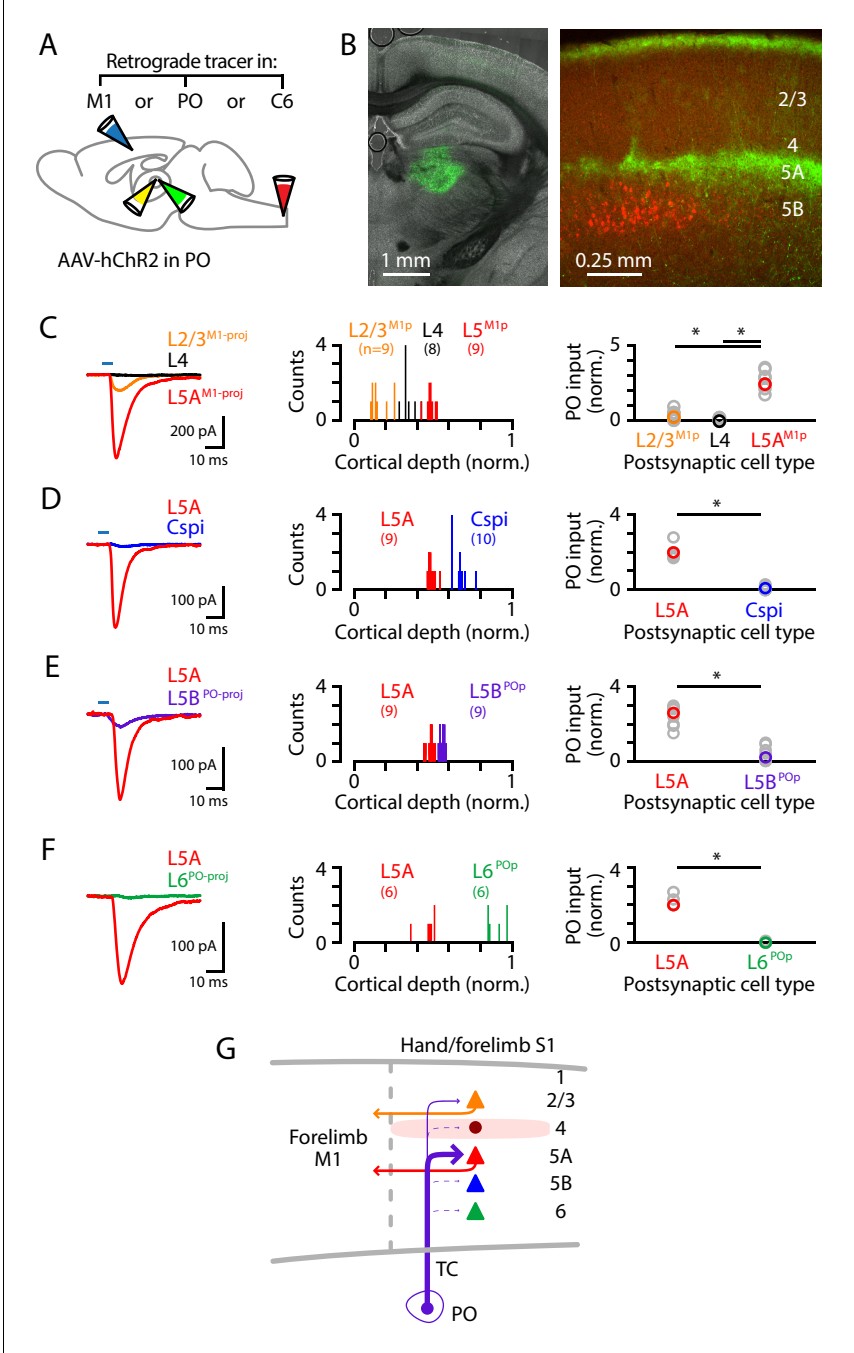

**Figure 5.** PO axons mainly excite L5A[M1-proj] neurons in S1. (**A**) Schematic of injection strategy: the PO was injected with AAV-hChR2, and the forelimb M1, PO, and/or C6 spinal cord with retrograde tracer(s) (CTB647 and/or red Retrobeads). (**B**) Left: coronal section showing labeling at the injection site in PO (green). Right: coronal section showing labeled PO axons (green) ramifying primarily in L1 and L5A of S1, and also showing the retrogradely labeled corticospinal neurons (red). (**C**) Left: example traces of EPSCs evoked by photostimulating the ChR2-expressing PO axons, recorded in L2/3[M1-proj], L4, and L5[M1-proj] neurons in S1. Middle: Histogram of the normalized cortical depths of each of the S1 cell types sampled. Numbers of cells per group are given in parentheses below the cell type labels. Right: Plot of EPSC amplitudes recorded in the three types of postsynaptic S1 neurons. Asterisks (*) indicate significant differences between groups (details in main text). (**D**) Same, comparing PO inputs to L5A and corticospinal[C6-proj] neurons in S1. (**E**) Same, comparing PO inputs to L5A and L5B[PO-proj] neurons in S1. (**F**) Same, comparing PO inputs to L5A and L6[PO-proj] neurons in S1. (**G**) Schematic summary of the main findings.

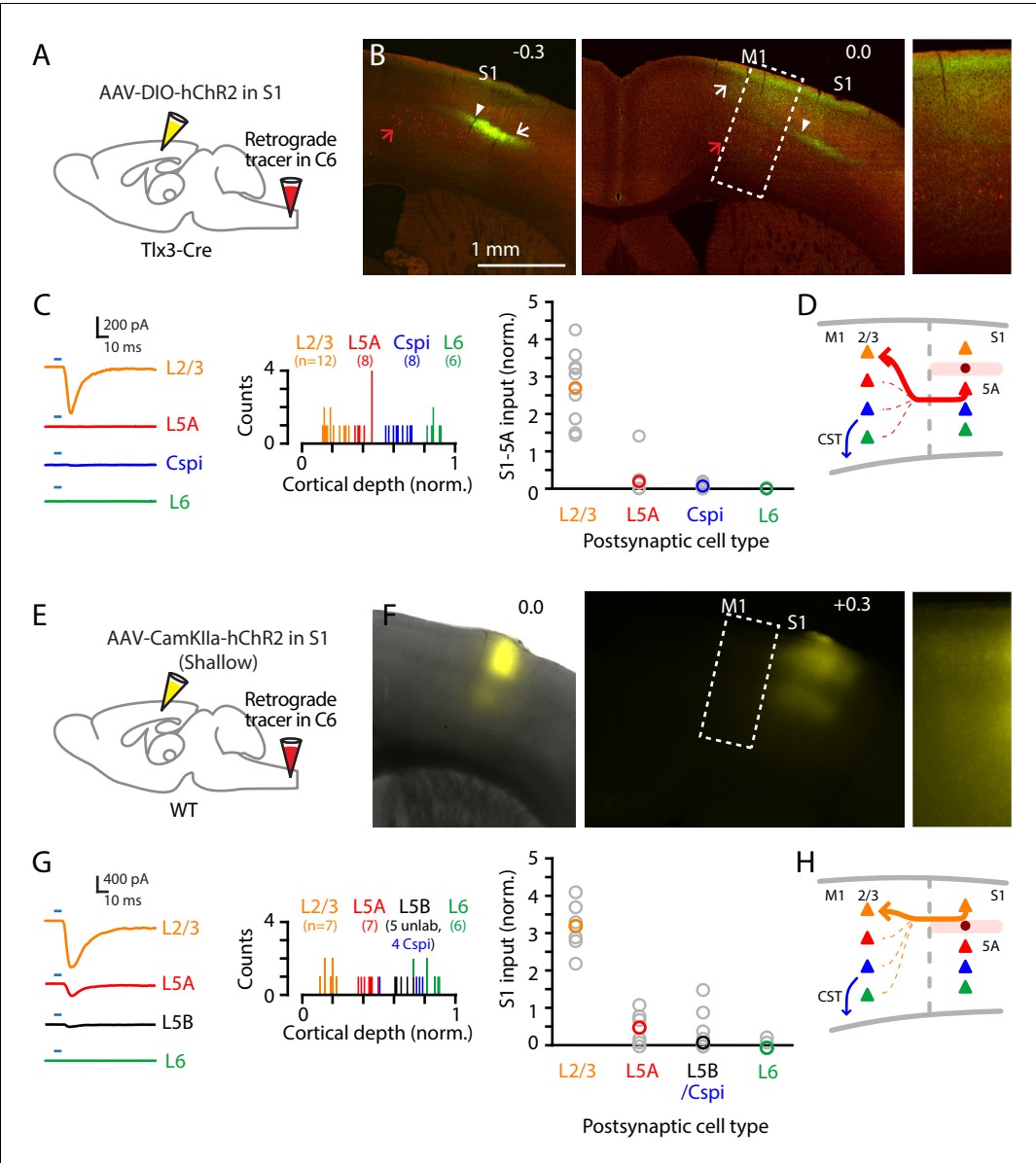

**Figure 6.** Corticocortical axons from S1 mainly excite L2/3 neurons in M1. (**A**) Schematic of injection strategy: the cervical spinal cord was injected at level C6 with retrograde tracer (red Retrobeads), and hand S1 was injected with AAV-DIO-hChR2, in a Tlx3-Cre mouse. (**B**) Left: Coronal section at the level of hand S1, showing labeling primarily of L5A neurons at the site of injection (arrow). Corticospinal neurons in L5B are also observed (red; red arrow). White arrowhead marks the approximate location of the medial border of hand S1. Center: Same, for a more anterior coronal section at the level of hand M1. Right: Same, showing an enlarged view of the labeling pattern in forelimb M1. (**C**) Left: Example traces of EPSCs evoked by photostimulating the ChR2-expressing S1 axons, recorded in L2/3, L5A, L6, and corticospinal[C6-proj] neurons in M1. Middle: Histogram of the normalized cortical depths of each of the S1 cell types sampled. Numbers of cells per group are given in parentheses below the cell type labels. Right: Plot of EPSC amplitudes recorded in the four types of postsynaptic M1 neurons. (**D**) Schematic summary of the main findings. (**E–H**) Same, but using shallow injections in S1 to label L2/3 neurons, to analyze the S1-L2/3→M1 connections.

## Discussion

Using multiple techniques for cell-type-specific dissection of circuit connections, we analyzed the excitatory synaptic connectivity along the somatosensory-to-motor, lemnisco-cortico-spinal, transcortical pathway that leads to and through the hand-related subfield of S1 and forelimb M1. In addition to

the current findings, prior results show that the L4 neurons in hand S1 strongly excite L2/3 neurons (*Yamawaki et al., 2014*), and the L2/3 neurons in forelimb M1 strongly excite cervically projecting corticospinal neurons (*Anderson et al., 2010*). Collectively, these results suggest a wiring diagram for the circuit architecture of the feedforward excitatory connections constituting a transcortical circuit for the mouse's hand and forelimb (*Figure 7*). A salient feature is the sharp contrast between the 'streamlined' organization of the lemnisco-cortical leg of the circuit, spanning the relatively large ~1 cm distance from cuneate to cortex via a single excitatory synapse in thalamus, and the densely polysynaptic organization of the corticocortical leg of the circuit, linking S1 to M1 across a mere ~1 mm distance but through complex circuits that engage multiple subtypes of intratelencephalic (IT) neurons (in L2 through L5A in S1, and in L2/3 in M1) en route to the M1 corticospinal neurons that close the transcortical loop by feeding into spinal circuits controlling motor neurons innervating forelimb muscles.

## Technical considerations

The circuit-analysis techniques used here each have certain advantages and limitations. For example, one tool we used was the recently developed Cre-dependent PRV-Introvert-GFP virus, which together with Cre-driver mouse lines enables cell types of interest to be selectively labeled as starter cells for polysynaptic circuit tracing (*Pomeranz et al., 2017*). General considerations with viral circuit-tracing methods include the possibilities of mixed neuronal tropism, under-labeling of connected neurons, and transsynaptic versus transneuronal propagation modes (*Luo et al., 2018*; *Beier, 2019*; *Nectow and Nestler, 2020*; *Rogers and Beier, 2021*). As discussed earlier (see Results), additional considerations with the use of PRV-Introvert-GFP for cortical labeling include the increasing difficulty over time of distinguishing first-, second-, and higher-order labeling in the cortex, due to interconnections among cortical neurons. However, this was not a major concern for our

main purpose of using PRV to anatomically delineate the lemnisco-cortical pathway leading to forelimb S1, as a starting point for subsequent detailed quantitative analysis of synaptic connectivity in this circuit using electrophysiology-based methods. In particular, we used the technique of ChR2-based circuit mapping, which combines selective presynaptic photostimulation and targeted postsynaptic whole-cell recordings. A limitation with this technique is that it gives only one particular (albeit particularly important) view of connectivity from the perspective of single-cell measurements at the soma (*Yamawaki et al., 2016*). Because the strengths and drawbacks of these techniques tend to be distinct and often complementary, the use of multiple techniques helps to establish findings by triangulation. Accordingly, we assessed connectivity along the transcortical circuit using several approaches, including anatomical labeling, circuit tracing with PRV, and anterograde labeling of axons with ChR2 and electrophysiological recordings from retrogradely labeled projection neurons.

We also developed a circuit analysis paradigm that combines ChR2-electrophysiology and virally mediated anterograde transneuronal labeling using AAV-hSyn-Cre (*Zingg et al., 2017*). With this approach, by starting at the cuneate and injecting multiple retrograde tracers to label various types of cortical projection neurons, we were able to sample and compare, in the same slices, cuneo-thalamo-cortical inputs to

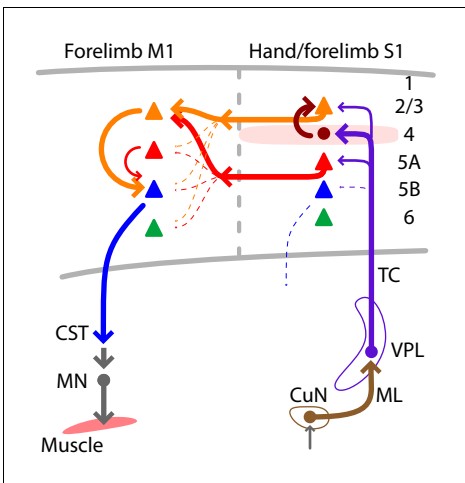

**Figure 7.** Summary wiring diagram of the major excitatory connections along the hand/forelimb-related somatosensory-to-motor transcortical circuit. The thickest arrows emphasize the strongest connections. The lemnisco-cortical circuit, arising from the cuneate nucleus, traverses the VPL via strong, depressing-type excitatory connections, and primarily targets L4 neurons in hand-related S1. In hand S1, similar to other sensory areas, L4 neurons connect strongly to L2/3 neurons. Neurons in both L2/3 and L5A in turn project to M1, forming convergent excitatory connections onto L2/3 neurons there. Strong local L2/3 connections to corticospinal neurons form the last connection to close the circuit leading back to the cervical spinal cord and the motor neurons controlling the forelimb musculature.

various corticocortical and corticospinal projection neurons, in effect constituting cuneo-thalamo-cortico-cortical and cuneo-thalamo-cortico-spinal circuits. This paradigm thus extends the number of circuit nodes that can be tested in the same experiment, from two, as in standard ChR2-based approaches (*Petreanu et al., 2007*), or three, as in approaches involving recordings from identified projection neurons (*Yamawaki et al., 2016*), to four, by selectively activating inputs from presynaptic neurons that are postsynaptic to a particular upstream source of interest.

## Areal organization of the hand/forelimb subfield of S1

Characterization of the areal topography of the hand/forelimb subfield of mouse S1 was aided by the Scnn1a-Cre mouse line, which labels L4 across all of S1, similar to cytochrome oxidase staining (*Woolsey and Van der Loos, 1970*; *Sigl-Glöckner et al., 2019*). Based on comparison to prior somatotopic mapping studies in other mammalian species and rats in particular (*Dawson and Killackey, 1987*; *Waters et al., 1995*), this anatomical characterization provides a working framework for the somatotopic layout of the mouse's hand representation in S1, with the hypothenar/ulnar aspect most medial, adjacent to the hindlimb subfield of S1, and the thumb/thenar region most lateral, adjacent to the lower lip subfield of S1.

Elucidation of this topography helps to further clarify the longstanding issue of the apparent overlap of S1 and M1 in the limb representations of rodent 'sensorimotor' cortex (*Hall and Lindholm, 1974*; *Donoghue and Wise, 1982*; *Frost et al., 2000*). For mouse forelimb S1, this overlap is evident anatomically as a zone where the distribution of cervically projecting corticospinal neurons extends beyond M1 into S1, and functionally as partly co-extensive motor and somatosensory maps (*Li and Waters, 1991*; *Ayling et al., 2009*; *Tennant et al., 2011*). Recent evidence reveals that corticospinal neurons in S1 in the overlap zone project to more dorsal levels of the spinal cord where they innervate distinct classes of spinal interneurons and are involved in more sensory-related aspects of forelimb motor behavior, whereas M1 corticospinal neurons project to more ventral cord levels and connect to spinal interneurons that directly contact spinal motor neurons (*Ueno et al., 2018*). Here, we confirmed that the cortical distribution of cervically projecting corticospinal neurons extends partially into hand S1, specifically along the medial aspect corresponding to the hypothenar/ulnar subregion.

This topography of the forelimb-related S1, adjacent to M1 and sharing the cortical distribution of corticospinal neurons, contrasts with that of the whisker-related areas in mouse cortex, where vibrissal S1 and M1 are widely separated as non-adjacent cortical areas. In this sense, forelimb S1 and M1 in the mouse resembles more the typical side-by-side topographic layout of somatosensory and motor areas seen in primates. Another notable aspect of the somatotopic layout of the forelimb subfield of mouse S1 is the relative expansion, or cortical magnification, both of the hand with respect to the rest of the forelimb, and of the thumb/thenar representation with respect to the rest of the hand. This pattern has been observed in other mammalian species, ranging from other rodents such as rats and squirrels to other primates and humans (*Sur et al., 1978*; *Waters et al., 1995*; *Jain et al., 2008*; *Krubitzer et al., 2011*; *Martuzzi et al., 2014*).

## Cuneothalamic connections

Photostimulation of cuneothalamic axons generated strong, depressing EPSCs in VPL neurons, as expected for ascending inputs to first-order sensory thalamic nuclei with 'driver' type inputs (*Sherman and Guillery, 1998*) and consistent with observations for lemniscal inputs to VPM (*Mo et al., 2017*). Cuneate axons did not excite all VPL neurons tested, despite the recorded neurons being within the fluorescently labeled axonal field. Most likely, given the fine-scale somatotopic organization of both the cuneate and VPL nuclei (*Li et al., 2012*; *Li et al., 2014*), this was because many axons were myelinated, en route to their topographically precise terminal arborizations within the VPL.

The cuneo-VPL projection and its whisker-related trigemino-VPM counterpart constitute the medial lemniscal pathway to somatosensory thalamus. In the whisker-barrel system, in addition to lemniscal inputs, thalamus receives paralemniscal afferents, which arise from other regions of the trigeminal nucleus (pars interpolaris, rostral subdivision) and innervate the PO nucleus. Thus, our retrograde tracer injections in the PO resulted in labeling in the trigeminal nucleus, but not in the cuneate nucleus. The PRV labeling using L5A neurons in hand S1 as starter neurons labeled the PO,

but also did not lead to additional labeling in the cuneate. Thus, for the hand-related pathways, and in contrast to cuneo-PO projections described in other species such as the cat (*Berkley et al., 1986*; *Loutit et al., 2021*), we did not identify a clear cuneo-PO counterpart to the trigemino-PO circuit in the whisker-related paralemniscal pathway. Although ascending subcortical sources of input to hand-related PO neurons in the mouse remain to be identified, descending cortical axons from hand S1 target a subregion of PO, strongly exciting recurrently projecting PO neurons there to form cortico-thalamo-cortical loops (*Guo et al., 2020*).

## Thalamocortical and corticocortical connections

The pattern of VPL connectivity to excitatory neurons in hand S1, marked by a strong bias toward L4 neurons, matched the anatomical pattern of axon branching, and accords generally with prior results in whisker-related pathways and core-type thalamocortical projections (*Petreanu et al., 2009*; *Cruikshank et al., 2010*; *Wimmer et al., 2010*; *Harris and Mrsic-Flogel, 2013*; *Adesnik and Naka, 2018*; *Sermet et al., 2019*). Our L4 recordings were targeted to putative excitatory neurons (based on the typically small soma size of S1 spiny stellate neurons) and thus leave unaddressed the patterns of VPL connections to diverse other excitatory and inhibitory neuron types in S1 L4 (*Staiger and Petersen, 2021*). The VPL input was moderately strong to L2/3 and L5A neurons, but weak or absent to corticospinal neurons. The pattern of PO inputs was distinct insofar as input was strong to L5A and weak to L4 neurons, but also similar in that input was again weak or absent to corticospinal neurons. These findings resemble prior findings for VPM and PO input to neurons in whisker S1 (*Bureau et al., 2006*; *Petreanu et al., 2009*; *Audette et al., 2018*; *Sermet et al., 2019*). Thus for both thalamocortical projections to hand S1, the major targets are intratelencephalic-type neurons in the upper and middle layers.

Although we found evidence for direct, monosynaptic corticocortical continuation of the transcortical circuit in the form of VPL inputs to L2/3[M1-proj] and L5A[M1-proj] neurons, these connections were only moderately strong and often much weaker than the inputs to L4 neurons. Strong local L4→L2/3 connectivity has been demonstrated in hand-related S1 of the mouse (*Yamawaki et al., 2014*), implying that the disynaptic VPL→L4→L2/3[M1-proj] circuit is a major route for excitatory signaling along the transcortical VPL→S1→M1 pathway. The S1→M1 corticocortical pathway originates mainly from L2/3[M1-proj] and L5A[M1-proj] neurons in S1. The axons of these neurons project to upper layers of M1 and innervate L2/3 neurons in particular, with notably scarce/weak connectivity to other types of neurons including corticospinal neurons. However, corticospinal neurons in forelimb M1 of the mouse receive particularly strong local excitatory input from L2/3 (*Anderson et al., 2010*). Thus, the present findings, together with prior circuit-mapping results, imply that local L2/3 neurons in M1 are the critical penultimate excitatory link in this transcortical circuit, postsynaptic to S1 corticocortical axons and presynaptic to M1 corticospinal neurons (*Figure 7*). A similar organization is implied for S1 corticospinal neurons, except that their L2/3 inputs can arise locally without intervening corticocortical circuits, as indicated by recent anatomical tracing studies (*Frezel et al., 2020*). It is important to note that while we emphasize here the mono- and disynaptic connections along the feedforward circuits, recurrent connections within and across cell classes in the circuit presumably generate complex, polysynaptically propagating activity patterns in vivo.

The overall thalamocortical-corticocortical circuit architecture in many ways closely resembles the corresponding whisker-related S1→M1 circuits, which have mostly been studied piece-wise but also involve the concatenation of excitatory connections. These include connections from VPM neurons mainly to L4 neurons in S1, from those mainly to L2/3 and other local neurons in S1, and from those to mainly L2/3 neurons in M1, which excite local L5B neurons including pyramidal-tract type neurons (*Farkas et al., 1999*; *Hoffer et al., 2003*; *Lefort et al., 2009*; *Petreanu et al., 2009*; *Aronoff et al., 2010*; *Hooks et al., 2011*; *Mao et al., 2011*; *Hooks et al., 2013*; *Yamashita et al., 2018*; *Sermet et al., 2019*). One apparent difference in the hand-related circuits (in addition to the apparent lack of an ascending paralemniscal pathway to PO, discussed above) is that VPL inputs are notably weak to corticospinal neurons, representing a major subtype of pyramidal-tract type neurons in L5B, whereas VPM inputs to L5B neurons in whisker S1 appear relatively stronger and are implicated in early cortical processing of somatosensory signals (*Petreanu et al., 2009*; *Wimmer et al., 2010*; *Constantinople and Bruno, 2013*; *Rah et al., 2013*; *Sermet et al., 2019*; *Egger et al., 2020*).

## Transcortical and cortico-thalamo-cortical circuits

A recent analysis of the cortico-thalamo-cortical circuit organization of hand-related S1 in the mouse indicates that these circuits tend to form strongly recurrent loops, with cortical axons strongly exciting recurrently projecting thalamocortical neurons in both VPL and PO (*Guo et al., 2020*). The present findings carry implications for understanding how transcortical and cortico-thalamo-cortical circuits intersect and interconnect, pointing to specific cell types and their connections whereby feedforward transcortical circuits are selectively integrated with recurrent loops between cortex and thalamus. As alluded to above, both the VPL and PO connections to S1 neurons were overwhelmingly biased toward neurons in layers 2/3, 4, and 5A, including M1-projecting corticocortical neurons in L2/3 and L5A. These neurons are all members of the broad class of intratelencephalic type neurons. In contrast, the thalamus-projecting neurons we recorded from in L5B and L6, representing subtypes of pyramidal-tract and corticothalamic type projection neurons, respectively, generally received little or no direct excitatory input from either thalamic nucleus, broadly consistent with previous findings in forelimb M1 (*Yamawaki and Shepherd, 2015*) and whisker S1 (*Petreanu et al., 2009*; *Crandall et al., 2017*; *Frandolig et al., 2019*; *Sermet et al., 2019*). Instead, their input likely includes strong local excitation from intratelencephalic neurons (*Lefort et al., 2009*; *Hooks et al., 2011*; *Hooks et al., 2013*; *Yamawaki and Shepherd, 2015*). Thus, the available evidence suggests that the feedforward thalamocortical circuits largely avoid direct innervation of thalamus-projecting neurons and instead engage mainly intratelencephalic type neurons, including subtypes involved either mainly in local excitatory circuits (L4 neurons) or in both local and corticocortical circuits (L2/3, L5A neurons). Particularly striking in this regard is the strong bias of PO inputs to L5A neurons, including L5A^M1-proj neurons, which thus appear as common elements shared by recurrent cortico-thalamo-cortical loops, local excitatory networks, and corticocortical circuits in the transcortical circuit. Another way to conceptualize this network is as an extended set of intersecting and selectively interconnecting looping circuits, within which the feedforward circuits constituting the transcortical circuit are fully embedded. This perspective dovetails with emerging concepts about the crucial role of looped circuit architecture for sensorimotor control of the forelimb (*Bizzi and Ajemian, 2020*; *Reschechtko and Pruszynski, 2020*).

## Comparing hand- and whisker-related transcortical circuits

Having made comparisons to the counterpart vibrissal somatosensory circuits along the way, here we highlight several key points. The overall picture to emerge is one of similarity. Both systems are, after all, part of the lemniscal pathway, the main ascending somatosensory system for tactile sensation: cuneo-VPL-cortical for the forelimb and trigemino-VPM-cortical for the vibrissae. The patterns of excitatory connections for the hand-related circuits (*Figure 7*) generally resemble those known for vibrissal circuits, both for the lemniscal VPM/VPL-related pathways and also the paralemniscal PO-related pathways (see references listed above; for recent reviews, see *Adesnik and Naka, 2018*; *Staiger and Petersen, 2021*; *Shepherd and Yamawaki, 2021*). As also noted earlier, a possible difference in the hand-related circuits is the notably weak-to-absent VPL input to corticospinal neurons, contrasting with evidence suggesting relatively stronger VPM input to similar neurons in vibrissal S1 (thick-tufted L5B pyramidal-tract type neurons; see references above). However, such comparisons must be tempered by various considerations including the possibility of basic differences in cellular composition and other properties of the two cortical areas; for example, vibrissal S1 and M1 are largely devoid corticospinal neurons. On the other hand, for the relatively short-distance corticocortical circuits studied here, it is striking that they adhere to the same pattern of targeting upper-layer neurons previously observed in the vibrissal S1-to-M1 projection, despite the greater distances bridged by the inter-areal projections in that pathway (*Mao et al., 2011*). Whether these conserved features of transcortical circuit connectivity hold for other somatosensory pathways as well (e.g. hindlimb, trunk, tongue) seems likely, a prediction that can be tested using similar approaches for cellular circuit analysis as applied here.

## Functional implications

The highly polysynaptic nature of the circuit organization at the cortical level suggests many possibilities for cellular mechanisms that may regulate and modulate the flow of excitation through the loop. These include inhibitory mechanisms, such as particular types of interneurons activated by

these circuits; for example, 'bottom-up' feedforward inhibition through S1 activation of fast spiking interneurons in M1 (*Murray and Keller, 2011*), and 'top-down' disinhibition through M1 activation of VIP[+] and somatostatin[+]interneurons in S1 (*Lee et al., 2013*). Indeed, an essential aspect of the concept of the transcortical pathway is that it is represents a key interface for integration of somatosensory, motor, and cognitive signals (*Conrad and Meyer-Lohmann, 1980*; *Evarts and Fromm, 1981*; *Evarts et al., 1984*; *Pruszynski and Scott, 2012*; *Reschechtko and Pruszynski, 2020*). Perhaps the dense incorporation of multiple types of IT neurons into this circuit increases its computational power by providing an expanded array of targets by which local and long-range inputs from diverse sources can modulate excitatory feedforward signaling along the connections feeding into M1 corticospinal neurons. Whereas this study focused on lemnisco-cortical pathways, which chiefly mediate forelimb tactile processing, an important related area for future research is the circuit organization of cuneo-cerebello-cortical pathways, which mediate forelimb proprioceptive processing and are also integrated and modulated at the cortical level (*Jörntell and Ekerot, 1999*; *Loutit et al., 2021*; *Reschechtko and Pruszynski, 2020*). With the many tools now available in mice for in vivo monitoring and modulation of specific cell types, the challenge will be to prioritize which cells and circuits to investigate in which behavioral paradigms. The characterization provided here of excitatory cell-type-specific connections in the somatosensory-to-motor transcortical circuit for the mouse's hand presents a framework for targeted investigation of how this circuit organization supports specific aspects of sensorimotor integration and forelimb tactile sensory perception and motor control.

# Materials and methods

## Key resources table

| Reagent type (species) or resource | Designation | Source or reference | Identifiers | Additional information |
|---|---|---|---|---|
| Strain, strain background (*M. musculus*) | Wild-type C57BL/6 | Jackson Laboratory | #000664; RRID:IMSR_JAX: 000664 | |
| Strain, strain background (*M. musculus*) | Scnn1a-Cre or B6;C3-Tg (Scnn1a-cre)3Aibs/J | Jackson Laboratory; (*Madisen et al., 2010*) | #009613; RRID:IMSR_JAX: 009613 | |
| Strain, strain background (*M. musculus*) | Txl3-Cre or B6.FVB(Cg)-Tg (Tlx3-cre)PL56Gsat/Mmucd | MMRRC; (*Gerfen et al., 2013*) | #041158-UCD; RRID: MMRRC_041158-UCD | |
| Strain, strain background (*M. musculus*) | CaMKII-Cre or B6.Cg-Tg (Camk2a-Cre)T29-1Stl/J | Jackson Laboratory; (*Tsien et al., 1996*) | #005359; RRID:IMSR_JAX: 005359 | |
| Strain, strain background (*M. musculus*) | Ai14 or B6.Cg-Gt(ROSA) 26Sor$^{tm14(CAG-tdTomato)Hze}$/J | Jackson Laboratory; (*Madisen et al., 2010*) | #007914; RRID:IMSR_JAX: 007914 | |
| Strain, strain background (*M. musculus*) | Ai96 or B6J.Cg-Gt(ROSA) 26Sor$^{tm96(CAG-GCaMP6s)Hze}$/MwarJ | Jackson Laboratory; (*Madisen et al., 2010*) | #028866; RRID:IMSR_JAX: 028866 | |
| Recombinant DNA reagent | AAV-ChR2-mCherry or AAV1.CamKIIa.hChR2 (E123T/T159C).mCherry. WPRE.hGH | Addgene | #35512; RRID:Addgene_ 35512 | |
| Recombinant DNA reagent | AAV-ChR2-Venus or AAV1. CAG.ChR2-Venus.WPRE. SV40 | Addgene | #35509; RRID:Addgene_ 35509 | |
| Recombinant DNA reagent | AAV-hSyn-Cre or AAV1. hSyn.Cre.WPRE.hGH | Addgene | #35509; RRID:Addgene_ 35509 | |
| Recombinant DNA reagent | AAVretro-GFP or AAV-CAG-GFP | Addgene | #37825; RRID:Addgene_ 37825 | |
| Recombinant DNA reagent | AAVretro-tdTomato or AAV-CAG-tdTomato | Addgene | #59462; RRID:Addgene_ 59462 | |
| Recombinant DNA reagent | PRV-EGFP or PRV-152 | CNNV; http://www.cnnv.pitt.edu | | |
| Recombinant DNA reagent | PRV-Introvert-GFP | J. Friedman; (*Pomeranz et al., 2017*) | | |

## Mice

Animal studies were approved by the Northwestern University Animal Care and Use Committee. In addition to wild-type (WT) C57BL/6 mice (Jackson), we used the lines listed in the Key Resources Table, all maintained on a C57BL/6 background. Expression patterns of these transgenic Cre lines have been described in the original papers cited and in the transgenic characterizations of the Allen Brain Institute, and are also further described in this study. As no sex-dependent differences were expected for the circuits to be studied, experiments were not explicitly designed to test for such differences. Mice were used as they became available, without selection based on sex. Overall, male and female mice were used in approximately equal numbers. No sex-dependent differences were found in sub-analyses of the data, and the data were accordingly pooled. Animals were housed with a 12 hr light/dark cycle and given free access to water and food. Mice were 1.5–3 months old at the time of the initial surgery and used in experiments 3–6 weeks later. Animal numbers for each type of experiment are given in the text and figures.

## Viruses and tracers

The adeno-associated viruses (AAV) and pseudorabies viruses (PRV) used are listed in the Key Resources Table. Standard PRV viruses were obtained from the Center for Neuroanatomy with Neurotropic Viruses (CNNV). The Cre-dependent PRV-Introvert (mCherry or GFP) was provided by Jeffrey Friedman (Rockefeller University) (*Pomeranz et al., 2017*). Retrograde tracers used in this study included red Retrobeads (Lumafluor) and cholera toxin subunit B conjugated with Alexa 647 (CTB647, Thermo Fisher).

PRV viruses were received from the indicated source and propagated on pig kidney epithelial cells (PK15). Stocks were harvested when cells displayed full cytopathological effect (2–3 days post-infection) and titered on PK15 cells. Prior to titering or use in animals, viral stocks were dispersed in a cuphorn sonicator at 100% amplitude for 10 cycles of 1.5 s 'on' followed by 1 s 'off'.

## Injections

Stereotaxic injections of AAV viruses and retrograde tracers were performed as previously described (*Yamawaki and Shepherd, 2015*; *Guo et al., 2018*). Briefly, mice were deeply anesthetized with isoflurane, placed in a stereotaxic frame, thermally supported, and given pre-operative analgesic coverage (0.3 mg/kg buprenorphine subcutaneously). Craniotomies were opened over the injection targets in the right hemisphere. Laminectomies were performed in the case of spinal injections at cervical level 6 (C6). Injection pipettes, fabricated from glass capillary micropipettes and beveled to a sharp edge, were loaded with virus or tracer solution by tip-filling and advanced to reach the stereotaxic target; injection volumes were 40–100 nL. Animals were post-operatively covered with analgesic (1.5 mg/kg meloxicam subcutaneously once every 24 hr for 2 days).

To determine optimal coordinates for the various anatomical structures targeted for injections in this study, we used standard atlases (*Dong, 2008*) as a starting point, and refined the targeting based on retrograde and anterograde labeling patterns. For example, for cuneate injections, based on retrograde labeling from VPL, we used coordinates of (in mm) anteroposterior (AP) –7.5 to –8.0, mediolateral (ML) +1.0 to+1.2, and dorsoventral depth (Z) –3.0 to –3.4. For thalamic injections, based on anterograde labeling from the cuneate and retrograde labeling from S1, we used coordinates for VPL of AP –1.9, ML +2.0, Z –3.7; and, for PO, AP –1.9, ML +1.2, Z –3.3. Based on a series of characterizations (described in the Results), forelimb S1 coordinates were AP 0.0, ML +2.4, Z –0.2 to –0.9 for D5, and AP +0.2, ML +2.7, Z –0.2 to –0.9 for D2; forelimb M1 coordinates were AP 0.0, ML +1.5, Z –0.2 to –0.9.

## PRV tracing

PRV injections were performed largely as described above, with minor modifications. Anesthesia was induced and maintained with ketamine (80–100 mg/kg) and xylazine (5–15 mg/kg). Three to five days (as indicated) after injection of PRV (10 nL), animals underwent intracardial perfusion-fixation with 4% paraformaldehyde (PFA) in PBS. The brain and spinal cord were harvested, cryosectioned (0.1 mm), and processed for immunohistochemical visualization of fluorescence labeling, as described previously (*Yamawaki et al., 2019*). To control for nonspecific spread of the virus, we injected the Cre-dependent PRV-Introvert-GFP into the cortex of wild-type mice; no labeling was

observed, consistent with the original characterization of this improved version of Cre-dependent PRV (*Pomeranz et al., 2017*).

## Cortical flat-mounts

Scnn1a-Cre;Ai14 mice, previously injected in the left C6 with AAVretro-GFP, were transcardially perfused with 4% PFA in PBS, and the brain was extracted and cut in half along the midline. The cortex in the right hemisphere was dissected free from the underlying white and gray matter structures, and the medial bank was gently unfolded to partially flatten the cortex. The tissue was placed in dish filled with PFA (4% in PBS), and gently compressed under a weighted glass slide overnight. The tissue was washed with PBS and sectioned to remove the upper and lower cortical layers, leaving a ~0.4 mm thick slice containing L4 and L5. The flattened slice was mounted on a slide and imaged on an epifluorescence microscope.

## Somatosensory mapping

Transcranial fluorescence imaging of somatosensory responses was performed as described previously (*Guo et al., 2020*), with several modifications. Briefly, after undergoing head-post mounting surgery, mice were injected with ketamine (80–100 mg/kg) and xylazine (5–15 mg/kg) and head-fixed under an epifluorescence microscope equipped with a blue LED (M470L2, Thorlabs), low-power objective lens (Olympus, XLFluor 2x/340, N/A 0.14), and monochrome camera (2048 × 1536 pixels, FS-U3-32S4M-C, FLIR Systems). The apparatus was mounted on a vibration isolation table and covered during experiments by a black enclosure.

Mice expressing GCaMP6s in L4 neurons of S1 (Scnn1a-Cre;Ai96) were used in experiments involving stimulation of different digits of the hand, and mice expressing GCaMP6s in all cortical excitatory neurons (CaMKII-Cre;Ai96) were used in experiments comparing hand and other body part representations. The stimulator consisted of a plastic probe affixed to a piezoelectric bimorph wafer (SMBA4510T05M, Steiner and Martin), controlled by linear driver (EPA-007–012, Piezo Systems). For stimulation of single digits, a thin metal probe (∅~0.5 mm, fashioned from a 27G needle by blunting its tip) was affixed to the plastic probe, and its tip was brought into position, just next to the digit (D2 or D5). For stimulation of different body parts, the plastic probe tip was positioned just next to, without contacting, the left hand (glabrous skin), hindpaw (glabrous skin), or lower jaw (hairy skin). Trials consisted of 5 s blue LED illumination, 4 s image acquisition (2 × 2 binning, 40 ms exposure, 17.4 dB, 20 fps), and 1 s stimulation (20 Hz sinusoidal command signal). Image acquisition and tactile stimulation began 1 and 3 s after LED onset, respectively. Stimulation trials were interleaved with no-stimulation trials (no command signal to the bimorph driver), and repeated 30 times (3 s inter-trial interval). Stimulus delivery and image acquisition were controlled by WaveSurfer (wavesurfer.janelia.org) through an NI USB-6229 data acquisition board (National Instruments). During the experiment, the mouse was thermally supported with feedback-controlled heating pad (Warner instrument). Bright-field images of the cranium were used to identify bregma. In some experiments with Scnn1a-Cre;Ai96 mice, spinal injections at C6 with AAVretro-tdTomato enabled imaging of corticospinal labeling as well.

For off-line data analysis, each frame was spatially binned by 8, and pre-stimulus baseline (20 frames) were averaged and subtracted from each frame to create ΔF/F images for each trial. Data for 'stimulation' and 'no stimulation' trials were grouped and averaged, and sensory maps were constructed by calculating the average value in the stimulus time window (20 frames) during stimulus trials and subtracting from this the corresponding average value in the same time window during 'no stimulation' trials. For the display, the maps from each animal were normalized, bregma aligned, averaged, and median filtered with a kernel of 5 × 5 pixels. Contours and centroids of responses were determined using 'regionprops' and other standard functions in Matlab.

## Circuit analysis

Methods for slice-based optogenetic-electrophysiological circuit analysis have been described in detail previously (*Yamawaki et al., 2019*). Briefly, mice that had undergone in vivo labeling were euthanized by isoflurane overdose and decapitation, and brains were rapidly removed and placed in chilled cutting solution (in mM: 110 choline chloride, 11.6 sodium L-ascorbate, 3.1 pyruvic acid, 25 $NaHCO_3$, 25 D-glucose, 2.5 KCl, 7 $MgCl_2$, 0.5 $CaCl_2$, 1.25 $NaH_2PO_4$; aerated with 95% $O_2$/5% $CO_2$).

Coronal slices (0.3 mm for cortex and 0.25 mm for thalamus) were cut (VT1200S; Leica) in chilled cutting solution, transferred to artificial cerebrospinal fluid (ACSF, in mM: 127 NaCl, 25 NaHCO$_3$, 25 D-glucose, 2.5 KCl, 1 MgCl$_2$, 2 CaCl$_2$, 1.25 NaH$_2$PO$_3$), incubated at 34 °C for 30 min, and kept at 22 °C for at least 1 hr prior to recording. Slices were placed in the recording chamber (perfused with ACSF at 32 °C) of a microscope equipped for whole-cell electrophysiology, photostimulation, and fluorescence microscopy. Ephus software (http://scanimage.vidriotechnologies.com/display/ephus/Ephus) (*Suter et al., 2010*) was used for hardware control and data acquisition. Recordings (serial resistance <40 MΩ) in voltage-clamp mode were made using borosilicate pipettes filled with cesium-based internal solution, composed of (in mM): 128 cesium methanesulfonate, 10 HEPES, 10 phosphocreatine, 4 MgCl$_2$, 4 ATP, 0.4 GTP, three ascorbate, 1 EGTA, 1 QX314; 4 mg/ml biocytin; pH 7.25, 290–295 mOsm. The sodium channel blocker QX-314 was included to prevent action potentials. Cortical recordings were made with TTX (1 µM) and 4-AP (100 µM) added to the ACSF, to isolate monosynaptic responses by blocking action potentials and thus polysynaptic inputs (*Petreanu et al., 2009*). These were omitted in thalamic recordings, where disynaptic responses were not a concern.

Wide-field photostimulation was performed using a low-power objective lens (4×) and a blue LED (M470L2; Thorlabs) driven by a TTL pulse to generate a 5 ms stimulus, with the LED intensity controller set to deliver 1 mW/mm$^2$ at the level of the specimen. For each cell, photostimulation trials were repeated several times at an inter-stimulus interval of 30 s, while recording in voltage-clamp mode with the command potential set to –70 mV. To quantify evoked synaptic responses, for each cell the traces from several (generally three) trial repetitions were averaged, and the response amplitude was calculated as the mean over a post-stimulus interval of 50 ms. Sequential recordings of multiple neurons were made for each slice. Data were compared by pooling across slices and animals, and pairwise comparisons were made using the absolute or normalized response amplitudes, as indicated in the text.

To test short-term synaptic plasticity, repetitive stimulation was performed by delivering short trains of photostimuli at 100 ms inter-stimulus interval. We initially attempted to use a laser-based approach but were limited by difficulty in activating axons at a location sufficiently far away from the recorded neuron, and therefore resorted to wide-field LED-based repetitive stimulation (*Jackman et al., 2014*). The response ratios were calculated based on the EPSC peak amplitudes, calculated by averaging five points around the peak responses.

### Experimental design and statistical analysis

Multiple animals were used per experiment, and results were analyzed by pooling across animals and/or by averaging per animal, as indicated in the Results. Group comparisons were made using non-parametric tests as indicated in the text, with significance defined as $p<0.05$. For two-group comparisons, the rank-sum test was used for unpaired data and the sign test for paired data. To compare three or more groups, the Kruskal-Wallis test was used. For group data, medians and median average deviations (m.a.d.) were calculated as descriptive statistical measures of central tendency and dispersion, except for ratios, for which geometric means and standard factors were calculated. No statistical methods were used to predetermine sample sizes, which are similar to those reported previously (*Yamawaki and Shepherd, 2015*). Statistical analyses were conducted using standard Matlab (Mathworks) functions.

## Acknowledgements

We thank John Barrett and Yutaka Yoshida for comments and suggestions, Frances Hausmann for technical assistance, and, for provision of PRV viruses, Jeffrey Friedman (Rockefeller), Oliver Huang (Princeton), Lisa Pomeranz (Rockefeller), and the Center for Neuroanatomy with Neurotropic Viruses (CNNV, University of Pittsburgh). Funding support was from NIH grants including NINDS R01 NS061963 (GMGS); NIAID R01 AI056346 (GAS); NIH Virus Center P40 OD010996 (CNNV).

## Additional information

### Competing interests
Gregory A Smith: Gregory A. Smith has disclosed a significant financial interest, Thyreos Inc.The Northwestern University Feinberg School of Medicine Conflict of Interest Review Committees has determined that this must be disclosed. The other authors declare that no competing interests exist.

### Funding

| Funder | Grant reference number | Author |
| --- | --- | --- |
| National Institutes of Health | NS061963 | Gordon M G Shepherd |
| NIAID | R01 AI056346 | Gregory Allan Smith |

The funders had no role in study design, data collection and interpretation, or the decision to submit the work for publication.

### Author contributions
Naoki Yamawaki, Conceptualization, Data curation, Formal analysis, Investigation, Visualization, Methodology, Writing - original draft, Project administration, Writing - review and editing; Martinna G Raineri Tapies, Data curation, Formal analysis, Investigation, Methodology; Austin Stults, Resources, Supervision, Investigation, Methodology; Gregory A Smith, Resources, Supervision, Funding acquisition, Methodology; Gordon MG Shepherd, Conceptualization, Resources, Supervision, Writing - original draft, Project administration, Writing - review and editing

### Author ORCIDs
Gregory A Smith (ID) https://orcid.org/0000-0001-9644-8472
Gordon MG Shepherd (ID) https://orcid.org/0000-0002-1455-8262

### Ethics
Animal experimentation: This study was performed in strict accordance with the recommendations in the Guide for the Care and Use of Laboratory Animals of the National Institutes of Health. All of the animals were handled according to approved institutional animal care and use committee (IACUC) protocols of Northwestern University.

### Decision letter and Author response
Decision letter https://doi.org/10.7554/eLife.66836.sa1
Author response https://doi.org/10.7554/eLife.66836.sa2

## Additional files

### Supplementary files
• Source data 1. Electrophysiology traces for the recordings shown in *Figures 3–6* are provided in a set of Matlab figure files.

• Transparent reporting form

### Data availability
All data generated or analysed during this study are included in the manuscript and supporting files (supplemental zip file).

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
