## [Decision Letter]

**Acceptance summary:**

Sensorimotor integration is central to the accurate execution of volitional movement. The mouse has emerged as a model for better understanding the circuitry underlying sensorimotor integration. Here, Yamawaki et al report results regarding the synaptic organization of ascending sensory pathways related to mouse forelimb somatosensory and motor cortex, shedding light on the circuits involved in limb sensorimotor integration.

**Decision letter after peer review:**

Thank you for submitting your article "Circuit organization of the excitatory sensorimotor loop through hand/forelimb S1 and M1" for consideration by *eLife*. Your article has been reviewed by 3 peer reviewers, one of whom is a member of our Board of Reviewing Editors, and the evaluation has been overseen Ronald Calabrese as the Senior Editor. The following individuals involved in review of your submission have agreed to reveal their identity: Barry Connors (Reviewer #2); Yutaka Yoshida (Reviewer #3).

Essential Revisions:

PRV tracing data:

If there is a principled relationship between time and the number of transsynaptic jumps, why are cortical neurons presynaptic to layer 4 (L4) neurons not seen after 72 hours incubation when presynaptic neurons in VPL are seen? Are some of the Cre expressing L4 neurons at 72 hours presynaptic to other infected L4 neurons or are they primary infections? And at 48 hours, are only L4 neurons labeled or also VPL neurons? In addition, in Figure 2D, the PRV labeling seems to be all layers in the S1. If starter (first order) neurons are detected in L4 48 hours after the infection, second order neurons would be detected 72 hours as shown in Figure 2B (VPL) and 3rd order neurons would be detected 96 hours after the infection. It is not clear that the labeling in the cortex and other brain regions is consistent with the time-course. How could synapse type or synapse number from a particular source affect these data and the interpretation? These issues should be placed in additional context in the Results and addressed in a revised Discussion.

Relatedly, how specific to Cre-expressing neurons is the PRV-invert-GFP labeling? It is hard to assess with the small example image in Figure 2B and the qualitative description of the control experiments in Cre- mice. Adding some quantification of these control experiments and higher magnification example images from Cre+ and Cre- mice would be helpful.

Finally, if the incubation goes even longer than 4 days, do labeled cells appear in PO?

Specification of tracers used:

The experiment in Figure 3, for example, used an anterograde tracer, but the type of tracer is never specified in the text or legend. Specifying precisely which retrograde tracers were used in which experiments in each figure would be helpful. If different combinations of tracers were used in an experiment, please specify which combinations contributed to which data.

In addition, it is difficult to discriminate the overlapping blue and green tracers in Figure 3F (right). Could something facilitate this discrimination such as adding a thin white line bordering the blue retrograde tracer (which seems to be a much smaller area than the green antero)?

Layer 4 cell types:

The experiment in Figure 4 is impressive, using multiple retrograde and anterograde viral tracers to selectively and transneuronally express ChR2 in cuneate-recipient VPL cells to dissect the concatenating Cun-VPL-S1 circuit. The presentation did raise a few questions.

First, the neurons in L4 are a mix of spiny excitatory cells and inhibitory interneurons, and in this experiment no genetic markers were used to tell them apart. In many thalamocortical projections (including VPM-to-L4 cells of S1) the inputs to PV interneurons are much stronger than those to spiny cells. Was any attempt made in this experiment to distinguish the neuronal types? If the cortical neurons were voltage-clamped with Cs-based solutions here, and spikes were blocked with TTX and QX-314, could that be clarified in the Methods (line 618) and Results? Furthermore, this issue of cellular diversity in L4 should be in the Discussion.

Second, though it is a small point, the thickness of the lines in schematic of Figure 4F exaggerate the differences in the data shown Figure 4E. For example, VPL inputs to L4 cells were, on average, ~2X those of the inputs to L2/3 cells, but the lines suggest a much larger difference. The picture also shows weak input to L6 pyramidal cells, but data from L6 cells were not reported in this experiment. Figure 7 also implies that VPL-to-L6 was studied.

Driver-type synapses:

The synaptic inputs from cuneate onto VPL neurons are described as "driver" type (line 197, Figure 3H,I). Apart from citing some Sherman papers, can the authors define what they themselves mean by driver connections? The value of the term as used here is not clear. Apart from a wide range of synaptic connection strengths, it is not obvious that there are any non-driver connections described in this paper. A related question: for many lemniscal-type inputs to thalamic neurons (notably trigeminal-to-VPm, but also retino-LGN), the connections from single axons are unusually strong. Were the authors able to estimate the size of unitary (presumed single-axon) responses in VPL cells by grading the optical stimulus intensity (i.e. measuring synaptic responses to minimal stimuli)?

Comparison with the whisker system:

The manuscript (rightfully and repeatedly) compares details of the forelimb circuits studied here with the central vibrissae circuits described in so many other studies. However, it takes a lot of careful reading through various Discussion sections to determine just what all of those differences and similarities are. The Discussion would benefit from one concise section (or paragraph, or even a table) that summarizes all the points of comparison and emphasizes, in particular, the differences between these two parallel systems.

Issue of partial overlap of limb representations in S1 and M1:

Line 344: Where the authors discuss the "longstanding issue" of partial overlap of limb representations in S1 and M1 of rodents, they cite Frost et al., 2000. This issue is much more longstanding than that. Relevant earlier references include Hall and Lindholm, Brain Res (1974), Wise and Jones, J Comp Neurol (1977), and Donoghue and Wise, J Comp. Neurol (1982). I'd suggest citing one or more of these earlier papers. In this study, Yamawaki et al. focus on the excitatory circuit organization of the lemnisco-cortical and corticocortical pathways which are important for the sensory-guided limb and hand control. To dissect those circuits, authors use sophisticated and combined techniques such as optogenetics, retrograde and anterograde AAVs, trans-synaptic viruses, electrophysiology, and mouse genetics. Using those impressive approaches, authors revealed the precise connectivity from the cuneate nucleus connects to the ventral posterior (VPL) thalamus, and then to the primary somatosensory (S1) and motor (M1) cortices. They also analyzed layer specific connectivity of those pathways.

Additional comments:

For Figure 1, AAV-GFP was injected into the spinal cord at C6. Can the authors provide more information about (even illustrate, perhaps) the extent and variability of the labeling at the site of the spinal injections?

Adding error bars as well as the individual traces in Figure 1C might be helpful as the lighter traces can be difficult to see.

In Figure 1D, there are spaces missing around "(n=4)" under corticospinal.

It would be helpful if more of the figures included orientation cues: lateral-medial (as in Figure 1A, but nowhere else in that figure), anterior-posterior, left or right hemisphere. Ideally these are indicated on the figure panels, but sidedness could be mentioned in the legends. Related: R and L in Figure 1 supplement 1D should be defined. The paper is anatomy figure-dense and many readers would benefit from more helpful labeling.

Do the authors have data from experiments similar to the one in Figure 2C using paw muscles? Such experiments would provide additional confirmatory evidence of the forelimb area in cortex.

In Figure 3, numbers of S1-projecting VPL neurons seems to be very low. Is this due to the low efficiency of the trans-synaptic anterograde AAV labeling? Or not many VPL neurons connect to the S1?

Figure 3 shows that only subsets (50%) of VPL neurons showed excitatory responses to cuneothalamic axon stimulation. How about cortical-VPL connections? Do all layer 4 neurons get excitatory inputs to the VPL axon stimulation in Figure 4?

In Figure 4B, AAV-hSyn-Cre and AAV-Flex-eGFP were injected in the cuneate nucleus, but the labeling was also detected in the gracile nucleus. Is this due to the leak of the injection? Or trans-synaptic labeling?

In Figure 4D (right panel), it is hard to distinguish the green signal from the cyan signal, although it is not impossible if viewing the figure on a monitor. Nonetheless, showing panels with each color separately in addition to the composite image may better show these data.

Line 249: "The anterogradely labeled PO axons ramified in L1 and L5A, as shown in an example from a Scnn1a-Cre x Ai14 mouse (Figure 5B)." This does not appear to be from a Scnn1a-Cre x Ai14 mouse. The red cells are corticospinal according to the legend.

Line 475: "into" is repeated twice in the line

Ai14 is described as an mCherry driver line in the table rather than a tdTomato reporter line.

---

## [Author Response]

Essential Revisions:PRV tracing data:If there is a principled relationship between time and the number of transsynaptic jumps, why are cortical neurons presynaptic to layer 4 (L4) neurons not seen after 72 hours incubation when presynaptic neurons in VPL are seen? Are some of the Cre expressing L4 neurons at 72 hours presynaptic to other infected L4 neurons or are they primary infections? And at 48 hours, are only L4 neurons labeled or also VPL neurons? In addition, in Figure 2D, the PRV labeling seems to be all layers in the S1. If starter (first order) neurons are detected in L4 48 hours after the infection, second order neurons would be detected 72 hours as shown in Figure 2B (VPL) and 3rd order neurons would be detected 96 hours after the infection. It is not clear that the labeling in the cortex and other brain regions is consistent with the time-course. How could synapse type or synapse number from a particular source affect these data and the interpretation? These issues should be placed in additional context in the Results and addressed in a revised Discussion.

These are good points and we have modified both the Results and Discussion sections to further address them. We do not have answers to all these questions. While in theory there may be a “principled relationship”, in practice there are multiple factors contributing to variability in the timing and extent of labeling. This limitation would certainly have been a major and indeed insurmountable problem had our purpose been to characterize the *cortical* labeling, because intracortical connectivity would have precluded accurate determination of synaptic order. But our only purpose was to label the ascending pathway from the cuneate. Because this is an essentially unidirectional pathway there is little cause for ambiguity about the synaptic order despite polysynaptic propagation and the possibility of variability in timing and other factors. We now make even more explicit some of the key limitations of the technique and how they do or don’t affect our results. We emphasize that the viral labeling was not taken as definitive proof of circuit connections, but provided a framework/starting-point for the subsequent detailed quantitative electrophysiology-based circuit analysis that constitutes the bulk of the study.

Relatedly, how specific to Cre-expressing neurons is the PRV-invert-GFP labeling? It is hard to assess with the small example image in Figure 2B and the qualitative description of the control experiments in Cre- mice. Adding some quantification of these control experiments and higher magnification example images from Cre+ and Cre- mice would be helpful.

Following the reviewers’ suggestion, we now show the images in an enlarged format in Figure 2—figure supplement 1, and have indicated the number of animals (n = 2). The sectioned material we have for these control experiments is unfortunately not ideal for accurately counting labeled neurons, but there were none that could be identified in the Cre-negative animals, whereas there were large numbers (tens to hundreds) in the Cre-positive animals. We note that the PRV-Introvert was generated by Pomeranz et al. (2017) based on a FLEx switch design, a non-leaky construct in wide use in Cre-dependent AAVs as well. We use the same PRV-Introvert as in the original paper by Pomeranz et al. They showed that this virus is essentially 100% Cre-dependent using multiple assays (gene expression, animal survival, etc.). Multiple subsequent studies have also confirmed the nonleakiness of Cre in this virus.

Finally, if the incubation goes even longer than 4 days, do labeled cells appear in PO?

We have tried this, but encountered two related problems: the mice began to showing behavioral abnormalities (e.g., tremor, turning), and the extensive neuronal labeling at this point in cortex and thalamus made it difficult to interpret the labeling patterns.

Specification of tracers used:The experiment in Figure 3, for example, used an anterograde tracer, but the type of tracer is never specified in the text or legend. Specifying precisely which retrograde tracers were used in which experiments in each figure would be helpful. If different combinations of tracers were used in an experiment, please specify which combinations contributed to which data.

We now specify the tracer used for each experiment in the figure legends, and labeled the schematics in several figures to indicate that retrograde tracers were used.

In addition, it is difficult to discriminate the overlapping blue and green tracers in Figure 3F (right). Could something facilitate this discrimination such as adding a thin white line bordering the blue retrograde tracer (which seems to be a much smaller area than the green antero)?

We now specify the tracer used for each experiment in the figure legends, and labeled the schematics in several figures to indicate that retrograde tracers were used.

Layer 4 cell types:The experiment in Figure 4 is impressive, using multiple retrograde and anterograde viral tracers to selectively and transneuronally express ChR2 in cuneate-recipient VPL cells to dissect the concatenating Cun-VPL-S1 circuit. The presentation did raise a few questions.First, the neurons in L4 are a mix of spiny excitatory cells and inhibitory interneurons, and in this experiment no genetic markers were used to tell them apart. In many thalamocortical projections (including VPM-to-L4 cells of S1) the inputs to PV interneurons are much stronger than those to spiny cells. Was any attempt made in this experiment to distinguish the neuronal types?

L4 recordings were aimed at putative spiny stellates, based on small soma size. We now comment on the cellular diversity of S1 L4 in the thalamocortical section of the Discussion.

If the cortical neurons were voltage-clamped with Cs-based solutions here, and spikes were blocked with TTX and QX-314, could that be clarified in the Methods (line 618) and Results? Furthermore, this issue of cellular diversity in L4 should be in the Discussion.

Clarifications added as suggested regarding the Cs-based internal, TTX, etc., in Results and Methods. As noted above, we comment on L4 cellular diversity in the Discussion.

Second, though it is a small point, the thickness of the lines in schematic of Figure 4F exaggerate the differences in the data shown Figure 4E. For example, VPL inputs to L4 cells were, on average, ~2X those of the inputs to L2/3 cells, but the lines suggest a much larger difference. The picture also shows weak input to L6 pyramidal cells, but data from L6 cells were not reported in this experiment. Figure 7 also implies that VPL-to-L6 was studied.

Changed as suggested, with line thickness of 2 points vs 1 point for inputs to L4 vs L2/3/5A, respectively. Lines to L6 cells have been removed in Figures 4 and 7.

Driver-type synapses:The synaptic inputs from cuneate onto VPL neurons are described as "driver" type (line 197, Figure 3H,I). Apart from citing some Sherman papers, can the authors define what they themselves mean by driver connections? The value of the term as used here is not clear. Apart from a wide range of synaptic connection strengths, it is not obvious that there are any non-driver connections described in this paper. A related question: for many lemniscal-type inputs to thalamic neurons (notably trigeminal-to-VPm, but also retino-LGN), the connections from single axons are unusually strong. Were the authors able to estimate the size of unitary (presumed single-axon) responses in VPL cells by grading the optical stimulus intensity (i.e. measuring synaptic responses to minimal stimuli)?

Good point; to better establish this as a driver-like synapse we would need to measure additional synaptic properties and compare to other synapses (e.g., CTVPL), which we have not done. We have edited the Results section to remove this descriptor.

Comparison with the whisker system:The manuscript (rightfully and repeatedly) compares details of the forelimb circuits studied here with the central vibrissae circuits described in so many other studies. However, it takes a lot of careful reading through various Discussion sections to determine just what all of those differences and similarities are. The Discussion would benefit from one concise section (or paragraph, or even a table) that summarizes all the points of comparison and emphasizes, in particular, the differences between these two parallel systems.

Point well taken. We have collected and expanded on some of these comparisons in a new stand-alone section in the Discussion, while trying to keep it concise and avoid redundancy. The lack of major differences is indeed the main message.

Issue of partial overlap of limb representations in S1 and M1:Line 344: Where the authors discuss the "longstanding issue" of partial overlap of limb representations in S1 and M1 of rodents, they cite Frost et al., 2000. This issue is much more longstanding than that. Relevant earlier references include Hall and Lindholm, Brain Res (1974), Wise and Jones, J Comp Neurol (1977), and Donoghue and Wise, J Comp. Neurol (1982). I'd suggest citing one or more of these earlier papers.

Excellent point; we now additionally cite Hall and Lindholm and Donoghue and Wise (but not Wise and Jones, which is less directly relevant).

In this study, Yamawaki et al. focus on the excitatory circuit organization of the lemnisco-cortical and corticocortical pathways which are important for the sensory-guided limb and hand control. To dissect those circuits, authors use sophisticated and combined techniques such as optogenetics, retrograde and anterograde AAVs, trans-synaptic viruses, electrophysiology, and mouse genetics. Using those impressive approaches, authors revealed the precise connectivity from the cuneate nucleus connects to the ventral posterior (VPL) thalamus, and then to the primary somatosensory (S1) and motor (M1) cortices. They also analyzed layer specific connectivity of those pathways.Additional comments:For Figure 1, AAV-GFP was injected into the spinal cord at C6. Can the authors provide more information about (even illustrate, perhaps) the extent and variability of the labeling at the site of the spinal injections?

Unfortunately, we did not quantify these parameters at the injection site in C6. We determined the success of injection based on the presence of labeling in the cortex. The cortical labeling was generally consistent across the mice.

Adding error bars as well as the individual traces in Figure 1C might be helpful as the lighter traces can be difficult to see.

To address the problem we have made the lighter traces darker and more visible.

In Figure 1D, there are spaces missing around "(n=4)" under corticospinal.

We are uncertain what the issue is.

It would be helpful if more of the figures included orientation cues: lateral-medial (as in Figure 1A, but nowhere else in that figure), anterior-posterior, left or right hemisphere. Ideally these are indicated on the figure panels, but sidedness could be mentioned in the legends. Related: R and L in Figure 1 supplement 1D should be defined. The paper is anatomy figure-dense and many readers would benefit from more helpful labeling.

We appreciate the suggestions, and have added more anterior/lateral indicators to various figure panels, as well as other clarifications.

Do the authors have data from experiments similar to the one in Figure 2C using paw muscles? Such experiments would provide additional confirmatory evidence of the forelimb area in cortex.

We tried this but were unable to achieve cortical labeling, with at least with 5 days incubation; the tiny size of the mouse’s hand muscles may be the problem.

In Figure 3, numbers of S1-projecting VPL neurons seems to be very low. Is this due to the low efficiency of the trans-synaptic anterograde AAV labeling? Or not many VPL neurons connect to the S1?

Is this referring to the seemingly low numbers of S1-projecting VPL that are seen in Figure 3F following retrograde tracer injection in S1? If so, this is a typical labeling pattern; focal injections of retrograde tracers in cortex result in focal labeling in subregions of thalamic nuclei, not nuclei in their entirety. In contrast, injecting the cuneate with AAV-GFP results in extensive labeling of cuneate (and to some extent also the nearby gracile) neurons, and thus extensive labeling of the cuneothalamic projection, as seen in Figure 3F. Or does this refer to Figure 4? If so, in that figure it can be seen that the anterograde transsynaptic labeling of S1-projecting VPL neurons is quite robust.

Figure 3 shows that only subsets (50%) of VPL neurons showed excitatory responses to cuneothalamic axon stimulation. How about cortical-VPL connections? Do all layer 4 neurons get excitatory inputs to the VPL axon stimulation in Figure 4?

Yes; essentially all L4 neurons received input (somewhat variable in amplitude). This is evident in the data for L4 input amplitudes shown in the plots in Figure 4E and Figure 4—figure supplement 1E,H. For these analyses of thalamic or cortical input to cortical neurons, we did not use event detection but instead relied on mean response amplitudes as our primary measure of connectivity.

In Figure 4B, AAV-hSyn-Cre and AAV-Flex-eGFP were injected in the cuneate nucleus, but the labeling was also detected in the gracile nucleus. Is this due to the leak of the injection? Or trans-synaptic labeling?

We now indicate (Figure 4B legend) that this may reflect leak/spread from the injection site.

In Figure 4D (right panel), it is hard to distinguish the green signal from the cyan signal, although it is not impossible if viewing the figure on a monitor. Nonetheless, showing panels with each color separately in addition to the composite image may better show these data.

Changes made as suggested; i.e., addition of the images showing each color separately.

Line 249: "The anterogradely labeled PO axons ramified in L1 and L5A, as shown in an example from a Scnn1a-Cre x Ai14 mouse (Figure 5B)." This does not appear to be from a Scnn1a-Cre x Ai14 mouse. The red cells are corticospinal according to the legend.

Fixed.

Line 475: "into" is repeated twice in the line

Fixed.

Ai14 is described as an mCherry driver line in the table rather than a tdTomato reporter line.

Fixed.